# Analysis of a Novel Fluidic Oscillator under Several Dimensional Modifications

**Kavoos Karimzadegan [1], Masoud Mirzaei [2] and Josep M. Bergada [1,*]**

[1] Fluid Mechanics Department, Universitat Politècnica de Catalunya, 08034 Barcelona, Spain; kavoos.karimzadegan@upc.edu
[2] Faculty of Aerospace engineering, K. N. Toosi University of Technology, Tehran 16569-83911, Iran; mirzaei@kntu.ac.ir
[*] Correspondence: josep.m.bergada@upc.edu; Tel.: +34-937-398-771

**Abstract:** To activate the boundary layer in Active Flow Control (AFC) applications, the use of pulsating flow has notable energy advantages over constant blowing/suction jet injections. For a given AFC application, five parameters, jet location and width, inclination angle, frequency of injection, and the momentum coefficient, need to be tuned. Presently, two main devices are capable of injecting pulsating flow with a momentum coefficient sufficient to delay the boundary layer separation: these are zero-net-mass-flow Actuators (ZNMFAs) and fluidic oscillators (FOs). In the present study, a novel FO configuration is analyzed for the first time at relatively high Reynolds numbers, and fluid is considered to be incompressible. After obtaining the typical linear correlation between the incoming Reynolds number and the outlet flow oscillating frequency, the effects of dimensional modifications on outlet width and mixing chamber wedge inclination angle are addressed. Modifications of the outlet width were observed to create large variations in FO performance. The origin of self-sustained oscillations is also analyzed in the present manuscript and greatly helps in clarifying the forces acting on the jet inside the mixing chamber. In fact, we can conclude by saying that the current FO configuration is pressure-driven, although the mass flow forces appear to be much more relevant than in previously studied FO configurations.

**Keywords:** fluidic oscillator design; computational fluid dynamics (CFD); flow control; feedback channel performance

## 1. Introduction

Among the wide range of applications that fluidic oscillators (FOs) can be employed in, it is relevant to highlight their use to enhance mixing [1,2], as heat transfer enhancers [3,4], as sensors to measure fluid flow [5–7], as fluidic sensors to measure micro/nanoscale transport properties [8], and as acoustic biosensors [9,10]. Perhaps the most common application is their use as Active Flow Control (AFC) devices to delay boundary layer separation on bluff bodies [11–13]. The use of pulsating flow in AFC applications provides the advantage of reducing the energy required to alter the boundary layer around bluff bodies. When considering options for producing pulsating jets, zero-net-mass flux actuators (ZNMFAs) and fluidic oscillators (FOs) stand out as promising choices. Notably, FOs have a distinct advantage because they rely on stationary components, enhancing their reliability, while the range of canonical shapes for FOs is somewhat restricted, delving into their performance becomes especially meaningful when tweaking internal dimensions. This involves examining variations in oscillation amplitude and frequency. The main objective of this paper is to shed light on this subject. In 2013, Bobusch et al. [14] carried out one of the initial assessments of fluidic oscillator performance by modifying its internal configuration. They provided recommendations concerning the inlet width of the mixing chamber to alter the output frequency of the fluidic actuator. Prior to this, in 2012, Vatsa et al. [15] investigated two different configurations of sweeping jet fluidic oscillators

(FO) using the lattice Boltzmann method and the PowerFLOW solver. Following this study, Ostermann et al. [16] conducted a more in-depth examination of these configurations in 2015. The two fluidic oscillators (FO) under examination bear similarities to those studied by Bobusch et al. [14] and Aram et al. [17], respectively. The velocity profiles generated by the FOs in quiescent air were compared with experimental data. The findings suggested that the FO with sharp internal corners, resembling the one employed in the study [14], produced a notably more consistent output velocity distribution in comparison to the oscillator with rounded internal corners. An analysis was conducted to compare the results of the two distinct setups, aiming to identify similarities and differences between the designs. Additionally, insights were provided into how these variations could potentially affect applications. Woszidlo et al. [18] examined a configuration that had been previously assessed by Gaertlein et al. [19]. Both of these configurations shared similarities with the one used by Bobusch et al. [14], with the main differences observed in the resulting output shape. In both [18] and [19], attention was centered on a single output. Indeed, Woszidlo et al. [18] directed their focus towards a thorough examination of flow phenomena within the mixing chamber and feedback channels. They observed that increasing the inlet width of the mixing chamber tended to raise the output frequency. Moreover, they discovered that introducing rounded features into the feedback channels led to a decrease in the formation of separation bubbles along these channels. Slupski and Kara [20] employed 2D-URANS simulations using Fluent software to investigate various geometry parameters for feedback channels (FC). The design of the sweeping jet actuator resembled the one examined by Aram et al. [17]. That study explored the impact of changing feedback channel height and width at different mass flow rates. All simulations were carried out under conditions of fully turbulent compressible flow using the SST k-omega turbulence model. The results indicated that oscillation frequencies increased with increases in the feedback channel height up to a certain threshold, beyond which they remained unchanged. In contrast, frequencies showed a decrease with the additional expansion of the feedback channel width. Wang et al. [21] carried out both experimental and numerical studies on a fluidic oscillator capable of producing frequencies across a wide range (50–300 Hz). Their investigation centered on examining the oscillation frequency response in relation to different lengths of the feedback channels. To accomplish this, they utilized 2D compressible simulations with the sonicFoam software and applied the k-epsilon turbulence model. Significantly, their results revealed a reverse linear correlation between frequency and the length of feedback loops. More precisely, decreasing the length of the feedback channel resulted in an increase in frequency. In 2018, Pandey and Kim [22] performed a three-dimensional numerical simulation using the SST turbulent model on the identical configuration previously employed by [14]. In this iteration, a single exit was employed, and the investigation was carried out at a Reynolds number of 30,000. Two geometric parameters, specifically the inlet and outlet widths of the mixing chamber, underwent adjustments. Significantly, modifying the inlet width had a significant influence on both the flow structure and the flow rate within the feedback channel, whereas minimal effects were noted when adjusting the outlet width. Investigating the impacts of changing the lengths of the feedback channel (FC) and the mixing chamber (MC) on output frequency and amplitude was performed by Seo et al. [23]; they utilized a 2D numerical model in 2018. The simulations were conducted assuming incompressible flow with a Reynolds number of 5000. Intriguingly, it was noted that elongating the feedback channel length did not bring about any alterations in the output frequency. This observation had been previously noted by [24], and both studies adopted incompressible flow assumptions, limiting the precision of the simulations in delivering precise information. In contrast, elongating the length of the mixing chamber resulted in a noticeable decrease in the actuator's output frequency. Baghaei and Bergada [25] developed a 3D simulation for a 3D fluidic oscillator. They implemented a comprehensive analysis on the forces driving the oscillations. In 2020, they used the same model and studied the effect of several design modifications [26]. Bergada et al. [27] studied the effect of feedback channel (FC) length on FO performance for

compressible flow conditions; they found that at large feedback channel lengths, the former main oscillation tends to disappear, the jet inside the mixing chamber simply fluctuates at high frequencies, and as the feedback channel (FC) length exceeds a certain threshold the FO stops oscillating. In Sarvar et al. [28], a novel shape of a fluidic oscillator (FO) in the laminar regime at a very low Reynolds number was studied; they observed that the jet sweeping angle amplitude is more pronounced for a two-dimensional FO as compared to a three-dimensional one at a fixed given Reynolds number, and the instability of the output jet becomes slightly chaotic at very low Reynolds numbers. In recent years, many researchers have been working on the analysis, design, and applications of FOs. For instance, Lee et al. [29], performed a numerical study of the influence of jet parameters of fluidic oscillator-type fuel injector on the mixing performance in a supersonic flow field. Their results showed that the influence of the sweeping jet angle on the mixing performance is more notable than that of the oscillating frequency, and they concluded that an appropriate combination of the frequency and sweeping jet angle is needed to maximize the mixing performance. In another attempt, Takavoli et al. [30] conducted a numerical investigation for enhancing a subsonic ejector performance by incorporating a fluidic oscillator as the primary nozzle. Their results indicated that a harmonically oscillating primary flow was generated, increasing the mixing entrainment and momentum transfer while reducing the pressure in the suction and mixing chambers. A microfluidic actuator was studied at high Reynolds numbers in [31], where 2D-URANS simulations along with experimental results and a final 3D CFD simulation was performed. Coherent structures as well as turbulent kinetic energy fields and velocity contours were used to unveil the associated complex fluid dynamics. The link between the output frequency and the feedback channel dimensional modifications was explored. CFD simulations using the k-omega (GECO) turbulence model with the aim of understanding thermal pollution in water jets was studied in [32]; they observed that the use of sweeping jet oscillators created a thermal dilution effect, drastically reducing the thermal pollution. The jet structures generated and their associated decay patterns were obtained. This effect was attributed to the fact that the sweeping jet converts more streamwise momentum into spanwise momentum, also increasing the homogeneity of the flow turbulence. Geometry modifications and scale variations of fluidic oscillators were considered in [33] to enhance mixing. The work was mostly performed via 2D-CFD simulations which were compared with 3D ones and experimental results. The most important design parameters were revealed, setting the basis of design modifications for other future applications. A supersonic fluidic oscillator employed to produce self-sustained oscillations in gas-pressurized chambers was studied in [34]. They observed that 2D-CFD simulations were not capable of reproducing the transients as accurately as the very computationally expensive 3D ones and decided to generate a low-order fluid circuit model which was quite capable of predicting the performance of the prototype.

In this study, a newly designed fluidic oscillator (FO), previously studied by [28] at very low Reynolds numbers is analyzed using numerical methods at relatively high Reynolds numbers, and flow is assumed to be incompressible. Following the establishment of a typical linear correlation between the Reynolds number and the oscillating frequency of the outlet flow, the study addresses the impact of dimensional modifications in outlet width and mixing chamber wedge inclination angle. Notably, alterations in the outlet width are observed to have a substantial influence on the performance of the fluidic oscillator. The manuscript also delves into an analysis of the origin of self-sustained oscillations, providing valuable insights into the forces acting on the jet within the mixing chamber. Ultimately, it is deduced that the current FO configuration operates under pressure-driven conditions, although mass flow forces appear to be much more significant than in previously studied FO configurations [25,26].

## 2. Governing Equations and Turbulence Model

In Computational Fluid Dynamics (CFD) simulations and when considering the fluid as incompressible, Navier–Stokes (NS) equations take the form:

$$\frac{\partial u_i}{\partial x_i} = 0 \tag{1}$$

$$\frac{\partial u_i}{\partial t} + \frac{\partial u_i u_j}{\partial x_j} = -\frac{1}{\rho}\frac{\partial p}{\partial x_i} + \nu\frac{\partial^2 u_i}{\partial x_j \partial x_j} \tag{2}$$

If the fluid is considered as turbulent, several approaches to solve NS equations are possible. Ideally, Direct Numerical Simulation (DNS) should be employed; the main problem associated with DNS is the extremely small mesh cells and time steps required. Then, the Kolmogorov length and time scales should be reached to properly detect the energy dissipation range. In the vast majority of the applications, the mentioned drawbacks make it impossible to use DNS simply because the computational time is far too large. The second approach is the use of Large Eddy Simulation (LES) as a turbulence model, it is a very good option in 3D flows, yet it still requires very fine meshes and large computational times. LES requires a Subgrid-Scale Model (SGS) which is used in areas where the vortical structures are smaller than the dimension of the mesh cells. Presently, LES turbulence modeling is extensively used although the computational time required, despite the fact that supercomputers are being used, is still too large for many applications. As a result, the vast majority of the nowadays CFD applications still need to be performed using Reynolds- Averaged Navier–Stokes (RANS) or URANS (unsteady RANS) turbulence models. Their precision is not as accurate as LES or DNS models, but the computational time needed shortens drastically, which allows the performance of 2D-CFD simulations with a reasonable degree of accuracy.

To discretize the NS equations under incompressible flow conditions, it needs to be kept in mind that the only variables associated with the NS equations are the pressure and the three velocity components. In order to be able to apply URANS models, each variable from the NS equations (we will call $\phi$ any generic variable) needs to be substituted by its average $\bar{\phi}$ and a fluctuation $\phi'$ term.

$$\phi = \bar{\phi} + \phi' \tag{3}$$

Once the mentioned substitution is performed, the resulting NS equations take the form (here just the equations for two-dimensional CFD models are presented):

$$\frac{\partial \rho}{\partial t} + \rho\left(\frac{\partial \bar{u}_x}{\partial x} + \frac{\partial \bar{u}_y}{\partial y}\right) = 0 \tag{4}$$

$$\frac{\partial \bar{u}_x}{\partial t} + \bar{u}_x\frac{\partial \bar{u}_x}{\partial x} + \bar{u}_y\frac{\partial \bar{u}_x}{\partial y} = -\frac{1}{\rho}\frac{\partial p}{\partial x} + g_x + \frac{1}{\rho}\frac{\partial}{\partial x}\left(2\mu\frac{\partial \bar{u}_x}{\partial x} - \rho(\bar{u}'_x)^2\right)$$
$$+ \frac{1}{\rho}\frac{\partial}{\partial y}\left(\mu\left(\frac{\partial \bar{u}_x}{\partial y} + \frac{\partial \bar{u}_y}{\partial x}\right) - \rho\overline{u'_x u'_y}\right) \tag{5}$$

$$\frac{\partial \bar{u}_y}{\partial t} + \bar{u}_x\frac{\partial \bar{u}_y}{\partial x} + \bar{u}_y\frac{\partial \bar{u}_y}{\partial y} = -\frac{1}{\rho}\frac{\partial p}{\partial y} + g_y + \frac{1}{\rho}\frac{\partial}{\partial x}\left(\mu\left(\frac{\partial \bar{u}_y}{\partial x} + \frac{\partial \bar{u}_x}{\partial y}\right) - \rho\overline{u'_y u'_x}\right)$$
$$+ \frac{1}{\rho}\frac{\partial}{\partial y}\left(2\mu\frac{\partial \bar{u}_y}{\partial y} - \rho(\bar{u}'_y)^2\right) \tag{6}$$

All RANS turbulence models are based on solving the Navier–Stokes equations by incorporating the concept of turbulence viscosity ($\mu_t$), which can be mathematically added

into the momentum equations described above using the subsequent differential definition (via the Boussinesq hypothesis):

$$\mu_t = \frac{-\rho \overline{u'_x u'_x}}{2\frac{\partial \bar{u}_x}{\partial x}} = \frac{-\rho \overline{u'_y u'_y}}{2\frac{\partial \bar{u}_y}{\partial y}} = \frac{-\rho \overline{u'_x u'_y}}{\frac{\partial \bar{u}_x}{\partial y} + \frac{\partial \bar{u}_y}{\partial x}} \tag{7}$$

In fact, the time-averaged fluctuation terms are usually grouped in a tensor called the apparent Reynolds stress tensor ($\tau_{app}$), which is a $3 \times 3$ symmetric matrix. Note that in the case of two-dimensional flow, this matrix explicitly consists of four terms.

The X and Y momentum terms of the NS equations can therefore be given as:

$$\frac{\partial \bar{u}_x}{\partial t} + \bar{u}_x \frac{\partial \bar{u}_x}{\partial x} + \bar{u}_y \frac{\partial \bar{u}_x}{\partial y} = g_x - \frac{1}{\rho}\frac{\partial p}{\partial x} + \frac{1}{\rho}\frac{\partial}{\partial x}\left(2(\mu + \mu_t)\frac{\partial \bar{u}_x}{\partial x}\right) + \frac{1}{\rho}\frac{\partial}{\partial y}\left((\mu + \mu_t)\left(\frac{\partial \bar{u}_x}{\partial y} + \frac{\partial \bar{u}_y}{\partial x}\right)\right) \tag{8}$$

$$\frac{\partial \bar{u}_y}{\partial t} + \bar{u}_x \frac{\partial \bar{u}_y}{\partial x} + \bar{u}_y \frac{\partial \bar{u}_y}{\partial y} = g_y - \frac{1}{\rho}\frac{\partial p}{\partial y} + \frac{1}{\rho}\frac{\partial}{\partial x}\left((\mu + \mu_t)\left(\frac{\partial \bar{u}_x}{\partial y} + \frac{\partial \bar{u}_y}{\partial x}\right)\right) + \frac{1}{\rho}\frac{\partial}{\partial y}\left(2(\mu + \mu_t)\frac{\partial \bar{u}_y}{\partial y}\right) \tag{9}$$

For the present study, we have decided to use the $k - \omega$ SST turbulence model, which relies on using the $k - \omega$ model near the wall, the $k - \epsilon$ model far away from the object, and a blending function between these two. For the chosen model, the turbulence viscosity $\mu_t$ can mathematically be given as:

$$\mu_t = \frac{\rho k}{\omega} \quad \longrightarrow \quad \begin{cases} \rho : \text{density} \\ k : \text{turbulent kinetic energy} \\ \omega : \text{turbulent kinetic energy specific dissipation rate} \end{cases}$$

This model employs two transport equations in order to solve $k$ and $\omega$. The equations used for each parameter are:

$$\frac{\partial k}{\partial t} + u_j \frac{\partial k}{\partial x_j} = P_k - \beta^* k\omega + \frac{\partial}{\partial x_j}\left[(\nu + \sigma_k \nu_T)\frac{\partial k}{\partial x_j}\right] \tag{10}$$

$$\frac{\partial \omega}{\partial t} + u_j \frac{\partial \omega}{\partial x_j} = \alpha S - \beta \omega^2 + \frac{\partial}{\partial x_j}\left[(\nu + \sigma_\omega \nu_T)\frac{\partial \omega}{\partial x_j}\right] + 2(1 - F_1)\sigma_{\omega 2}\frac{1}{\omega}\frac{\partial k}{\partial x_i}\frac{\partial \omega}{\partial x_i} \tag{11}$$

$F_1$ and $F_2$ are the blending functions which value depends on the distance of the cell to the wall. The first blending function takes the value of 0 far away from the wall, 1 in the cells close to the wall, and values between 0 and 1 in the transition region. The second blending function $F_2$ depends on the perpendicular distance from the wall (*d*) and, according to [35], since the modification to the eddy viscosity has its largest impact in the wake region of the boundary layer, it is imperative that $F_2$ extends further out into the boundary layer than $F_1$. Mathematically, it is expressed as:

$$F_2 = \tanh\left(\arg_2^2\right) \tag{12}$$

$$\arg_2^2 = \max\left(\frac{2k}{\beta^* \omega d}; \frac{500\nu}{\omega d^2}\right) \tag{13}$$

The constants of the arg function are adjusted manually, and for the present case, the values of the used turbulence model taken are: $\alpha_1 = 0.556$, $\alpha_2 = 0.44$, $\beta^* = 0.09$, $\beta_1 = 0.075$, $\beta_2 = 0.0828$ $\sigma_{k1} = 0.85$, $\sigma_{k2} = 1$, $\sigma_{\omega 1} = 0.5$, and $\sigma_{\omega 2} = 0.856$. Additionally, for more detailed information of the $k - \omega$ $SST$ turbulence model proposed, see [35].

## 3. Mesh Assessment

Figure 1 introduces the central part of the (FO) investigated in the present study. The mixing chamber (MC) is the core of the FO; above and below it, the feedback channels (FC) can be observed. Flow enters through the power nozzle (PN) and leaves through the main outlet. The red line shown in the mixing chamber upper converging wall is simply a line probe to allow the measurement of the maximum stagnation pressure in this location. Although not shown in the figure, the same line probe was also placed at the lower converging wall. The different dimensions of Figure 1 are as follows: FO power nozzle width 0.02 m, FO power nozzle angle $\theta = 8.343$ degrees, MC entrance 0.0255 m, MC output 0.08356 m, MC inlet angle $\alpha = 14.997$ degrees, FO outlet width 0.0266 m, FO outlet angle $\beta = 58.235$ degrees, external radius $Rext = 0.0949$ m, and internal radius $Rint = 0.0627$ m. The computational domain inlet width, which can be seen in Figure 2a, is 0.0808 m and the radius of the buffer zone, also seen in the same figure, is 1.562 m. This configuration is flat, having a spanwise length of 0.01 m; it is called the baseline case, and in the present manuscript, the main effect on the FO performance when modifying the FO outlet width and the MC inlet angle is analyzed. Figure 1 also shows the positive and negative directions taken for each of the two geometry modifications.

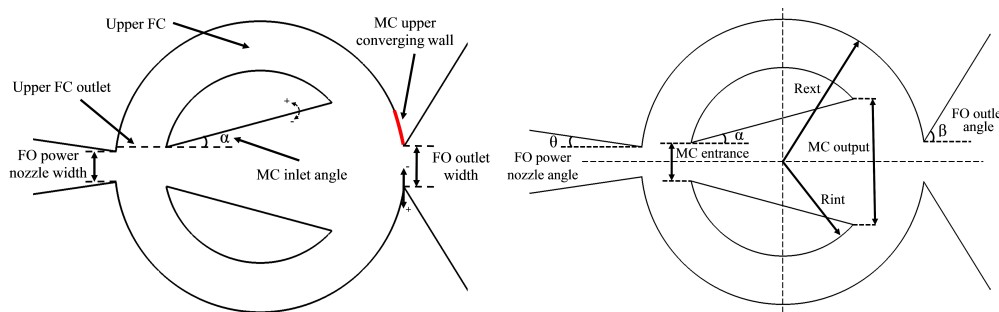

**Figure 1.** Fluidic oscillator mixing chamber main parameters.

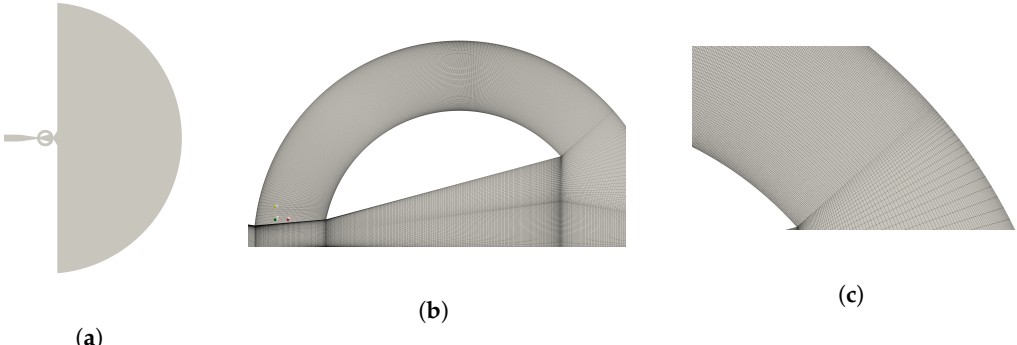

**Figure 2.** Main view of the computational domain used in the present study. (**a**) Computational domain overall view. (**b**) Mesh in the central part of the fluidic oscillator, upper view. (**c**) Upper feedback channel inlet, zoomed view of the mesh.

A fully structured two-dimensional (2D) mesh (extruded in the spanwise direction and using a single cell) with 367,720 cells was used to evaluate the flow at a Reynolds number Re = 54,595, the characteristic length being the fluidic oscillator power nozzle. In fact, in order to obtain simulated results independent of the mesh resolution, four different mesh densities were initially evaluated, their respective number of cells were, 62,143; 180,065; 367,720; and 718,920. Table 1 summarizes the minimum, maximum, and average $y+$ as well as the outlet mass flow frequency associated, obtained for the different mesh resolutions. The last column indicates the frequency error versus the frequency obtained using the denser mesh; notice that the error gathered using a mesh of 367,720 cells

is negligible. Figure 2a presents the entire computational domain with the mesh associated, and Figure 2b,c introduces a zoomed view of the upper feedback channel mesh.

The boundary conditions employed in all simulations were as follows: Dirichlet boundary conditions for velocity and Neumann boundary conditions for pressure at the inlet. An absolute pressure of $10^5$ Pa and Neumann boundary conditions for velocity were considered at the buffer zone outlet. Neumann boundary conditions for pressure and Dirichlet boundary conditions for velocity were set to all walls. This considers a turbulent intensity of $I = 0.1\%$, a fluid velocity of $U = 10$ m/s, and a computational domain inlet width of $L = 0.0808$ m (see Figure 2a). The following equations were used for the different turbulence properties: $k = (3/2)U^2I^2$; $\epsilon = C_\mu(k^{3/2}/L)$, where $C_\mu = 0.09$; $\omega = \epsilon/(C_\mu k)$; and $\nu_t = k/\omega$. Using those equations, values for the turbulent kinetic energy $k$, the turbulent dissipation rate $\epsilon$, the specific dissipation range $\omega$, and the turbulent kinematic viscosity $\nu_t$ at the computational domain inlet were, respectively, $k = 1.5 \times 10^{-4}$ (m$^2$/s$^2$), $\epsilon = 2.046 \times 10^{-6}$ (m$^2$/s$^3$), $\omega = 0.152$ (s$^{-1}$), and $\nu_t = 9.9 \times 10^{-4}$ (m$^2$/s). Table 2 summarizes the boundary conditions employed for the CFD simulations.

**Table 1.** Results from the mesh independence study.

| | | Mesh Assessment | | | |
|---|---|---|---|---|---|
| **2D Mesh Cells** | **Min y+** | **Average y+** | **Max y+** | **Frequency (Hz)** | **Error %** |
| 62,143 cells | 0.007893928 | 5.255150 | 63.09918 | 24.525 Hz | 15.37% |
| 180,065 cells | 0.5630300 | 1.616285 | 16.79723 | 29.181 Hz | 0.69% |
| 367,720 cells | 0.2748158 | 0.9271403 | 8.044681 | 29.091 Hz | 0.38% |
| 718,920 cells | 0.2741867 | 0.7549317 | 8.368831 | 28.98 Hz | - |

**Table 2.** Summary of the boundary conditions for the standard case (STD), also called the baseline case. Reynolds number = 54,595.

| | Boundary Conditions 2D-CFD | | | | |
|---|---|---|---|---|---|
| | **k (m$^2$/s$^2$)** | **Omega (s$^{-1}$)** | **Nut (m$^2$/s)** | **P (Pa)** | **U (m/s)** |
| inlet | $1.5 \cdot 10^{-4}$ | 0.152 | $9.89 \cdot 10^{-4}$ | zeroGradient | 10 |
| outlet | zeroGradient | zeroGradient | zeroGradient | $10^5$ | zeroGradient |
| Top & Bottom | empty | empty | empty | empty | empty |
| walls | $10^{-20}$ | omegaWallFunction value $= 10^{-5}$ | nutWallFunction value $= 10^{-7}$ | zeroGradient | 0 |

In the present study, the flow was considered as turbulent, incompressible, and isothermal, and all simulations were two-dimensional. The fluid used was air at 15 degrees Celsius, and the dynamic viscosity and the density were $\mu = 1.813 \times 10^{-5}$ kg/(m s) and $\rho = 1.225$ kg/m$^3$, respectively. The software OpenFOAM version 5.0 was employed for all simulations; the finite volume method is the approach OpenFOAM uses to discretize Navier–Stokes equations. Inlet turbulence intensity was set to 0.1% in all cases, pressure implicit with splitting operators (PISO) was used to solve the Navier–Stokes equations, the timestep being $4 \times 10^{-7}$ s, and spatial discretization was set to second order. Simulations were not considered to be converged until the residuals for the fluid velocity components, the turbulent kinetic energy, the specific dissipation rate, and the turbulent kinematic viscosity were to the order of $10^{-6}$ or $10^{-7}$. The residuals for the pressure were to the order of $10^{-4}$ after all simulations.

In order to further check the accuracy obtained with the different mesh densities, we decided to analyze some dynamic characteristics and introduce them in Figure 3. The FO outlet mass flow pulsating frequency is presented in Figure 3a. The instantaneous outlet mass flow and the lower feedback channel mass flow are shown in Figure 3b and

Figure 3c, respectively. The unsteady stagnation pressure measured at the mixing chamber lower converging wall is introduced in Figure 3d. Regardless of the graph chosen, it is observed that the dynamic results obtained with the meshes of 367,720 and 718,920 cells are almost identical, and clear differences are seen when using lower-resolution meshes. At this point, we can conclude that the mesh of 367,720 cells is perfectly good to perform the required simulations.

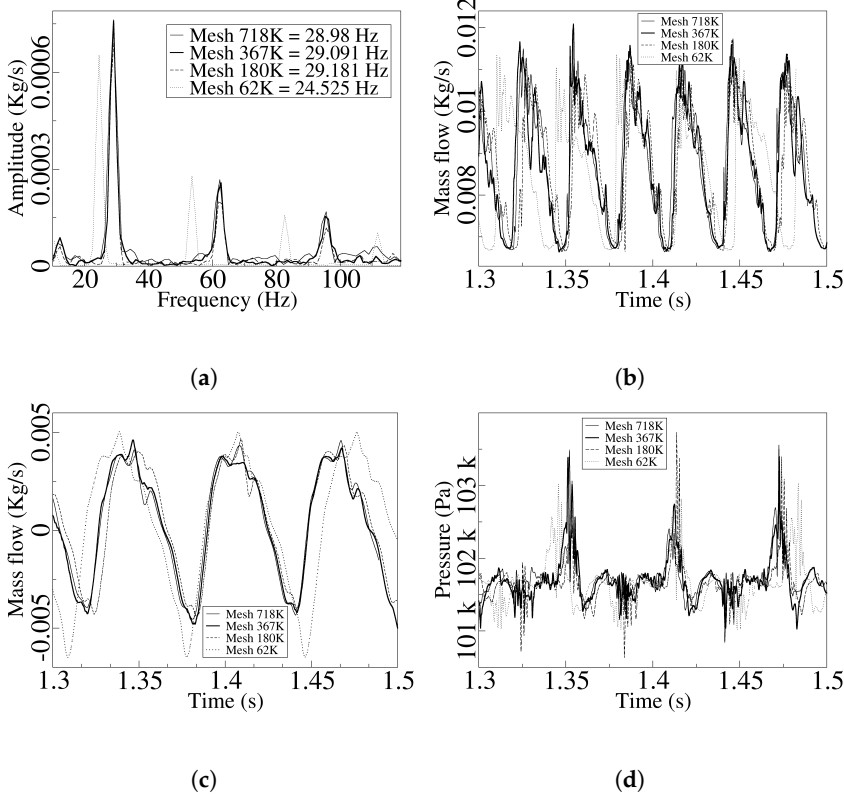

**Figure 3.** Fluidic oscillator performance at Reynolds number Re = 54,595 and for the four different mesh densities considered. Outlet mass flow frequency (**a**). Outlet mass flow (**b**). Lower feedback channel mass flow (**c**). Stagnation pressure measured at the mixing chamber downwards converging wall (**d**).

At this point, and in order to evaluate the effect of the inlet turbulence intensity on the results obtained, the mass flow at the FO outlet width and at the FC lower outlet were studied for three different FO inlet turbulence intensities, 0.01%, 0.1% and 1%; the results are presented in Figure 4. The mesh used had 367,720 cells and the Reynolds number was kept constant to *Re* = 54,595. As the turbulence intensity increases, the curves slightly drift towards one side, but minor differences are observed when comparing the evaluated minimum and maximum turbulence intensities. We can conclude that using a turbulence intensity of 0.1% at the FO inlet is perfectly correct. To further validate the results presented in this manuscript, we performed a full 3D simulation of the fluidic oscillator extruding the 2D mesh chosen a distance of 0.01 m in the spanwise direction, and using 21 cells, the resulting mesh had 7,722,120 cells (around 7.7 million cells). In order to be able to properly capture the boundary layer effects, the cells in the spanwise direction were concentrated near the walls, as observed in Figure 5. The minimum, maximum, and average Y+ after the 3D simulations were, respectively, $1.71 \times 10^{-3}$, 21.7, and 1.39, which certify the good quality of the mesh. The turbulence model, the inlet turbulence intensity, as well as the characteristic length were the same as the ones employed in the 2D simulations, and the Reynolds number considered was *Re* = 54,595. The boundary conditions were the same as the ones used in the 2D-CFD simulations, except for the top and bottom surfaces which in the 3D-CFD simulations were set to walls. Therefore, the boundary conditions presented in

Table 2 for the 2D simulations are applicable to the 3D ones simply by removing the entire (Top & bottom) line.

The comparison between the 2D and 3D simulations for the fluidic oscillator unsteady outlet mass flow, the unsteady stagnation pressure at the mixing chamber lower converging wall, and the frequency spectrum of the FO outlet mass flow signals are introduced in Figure 6a–c, respectively. The comparison shows very similar results in all the graphs presented, the main difference being the fluidic oscillator output frequency, which was about 11% higher for the 3D simulations. The conclusion from the present section is that the 2D-CFD simulations are reliable enough to be used as a tool to compare the FO performance when several Reynolds numbers and geometrical modifications are considered, which is what will be presented in the results section of the paper.

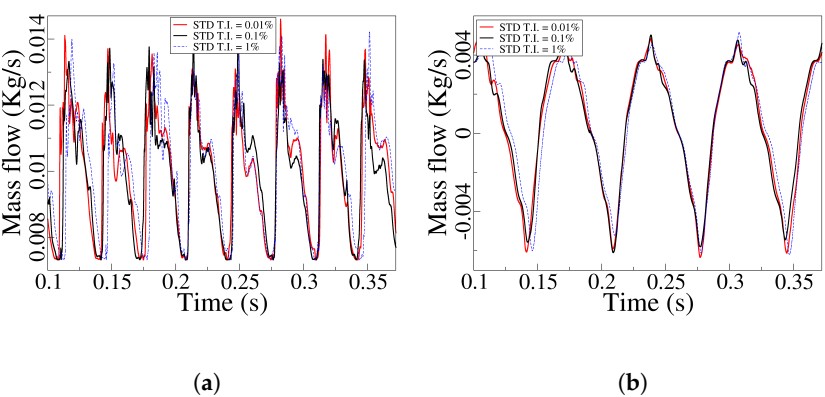

(**a**)    (**b**)

**Figure 4.** Fluidic oscillator outlet mass flow (**a**) and feedback channel lower outlet mass flow (**b**) for three different turbulence intensities (0.01%, 0.1%, and 1%) at the FO inlet. The standard (STD) Reynolds number 54,595 was used in all cases.

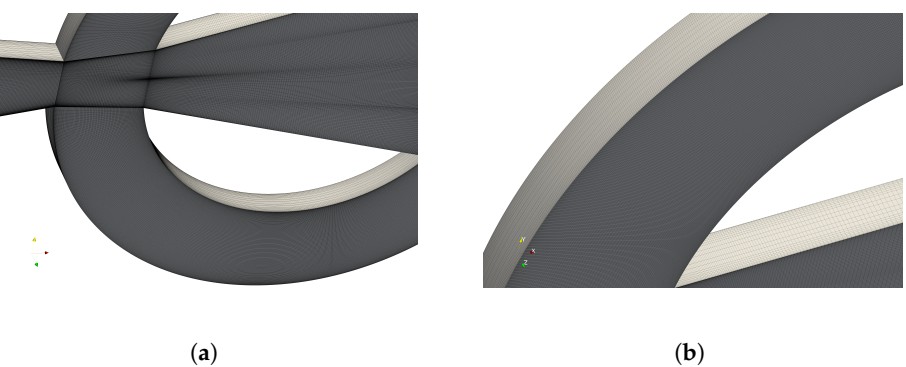

(**a**)    (**b**)

**Figure 5.** Three-dimensional mesh in the fluidic oscillator mixing chamber (**a**). Zoomed view of the 3D mesh in the upper feedback channel (**b**).

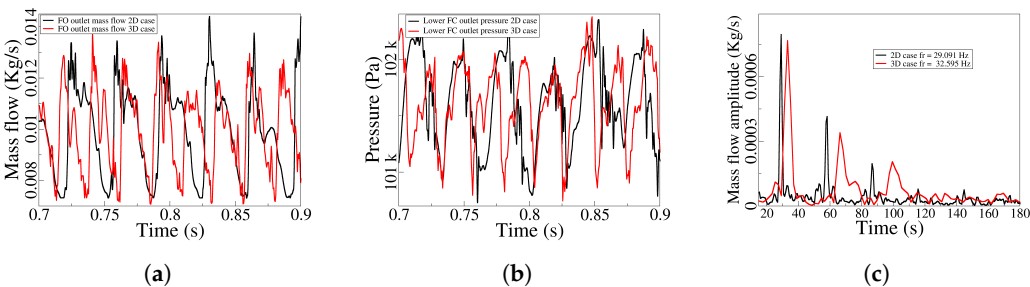

(**a**)    (**b**)    (**c**)

**Figure 6.** Fluidic oscillator outlet mass flow (**a**). Unsteady pressure at the lower feedback channel outlet (**b**). Fast Fourier transformation of the unsteady FO outlet mass flow (**c**). Comparison between 2D and 3D CFD simulations. The standard (STD) Reynolds number of 54,595 was used in all these cases.

## 4. Geometrical Modifications Considered

In the present study, a parametric analysis was performed to analyze the 2D-FO main characteristics under different geometrical modifications while keeping the Reynolds number constant at $Re = 54,595$. Two different modifications were considered, the mixing chamber outlet width and the mixing chamber inlet angle. Six different outlet widths ranging between 75% and 150% of the baseline outlet width and three inlet angles ranging between 79.61% and 122.55% of the baseline inlet angle were evaluated. Their different associated dimensions and percentages are presented in Tables 3 and 4, respectively.

**Table 3.** The different mixing chamber outlet widths considered in the present study.

| | Outlet Widths Evaluated | | | | | |
|---|---|---|---|---|---|---|
| Outlet width in % | $D_2 = 75\% D_4$ | $D_3 = 87.5\% D_4$ | $D_4 = (Baseline)$ | $D_5 = 112.5\% D_4$ | $D_6 = 125\% D_4$ | $D_7 = 150\% D_4$ |
| Outlet width in (m) | 0.02 | $0.0233\hat{3}$ | $0.02666\hat{6}$ | 0.03 | $0.0333\hat{3}$ | $0.03999\hat{9}$ |
| Outlet section name | 1 | 2 | $STD = 3$ | 4 | 5 | 6 |

**Table 4.** The different mixing chamber angles considered in the present study.

| | Mixing Chamber Inlet Angles | | |
|---|---|---|---|
| Inlet angle in % | $\alpha_1 = 79.61\% \alpha_2$ | $\alpha_2 = (Baseline)$ | $\alpha_3 = 122.55\% \alpha_2$ |
| Inlet angle in (degrees) | 11.9402 | 14.9968 | 18.3788 |

## 5. Concept of Momentum, Pressure, and Mass Flow Terms Acting on the Jet Entering the Mixing Chamber

To be able to evaluate the lateral forces pushing the jet as it enters the mixing chamber, the momentum (M) terms on the FCs' outlets are determined. Momentum is characterized by two terms: the pressure and the mass flow (notice that the units of each term are Newtons (N)). Both terms were instantaneously evaluated at the feedback channels upper and lower outlets, see Figure 1, the equation considered takes the form:

$$F = \dot{m}_{out} \times V_{out} + P_f \times S_{out} = \dot{m}_{out}^2 / (S_{out} \times \rho) + P_f \times S_{out} \tag{14}$$

where $\dot{m}_{out}, V_{out}, S_{out}$ and $P_f$, respectively, are the FC outlet instantaneous mass flow defined as $\dot{m} = \int_s \rho \vec{V} d\vec{s}$, the spatial-averaged fluid velocity, the FC outlet surface, and the spatial-averaged pressure instantaneously appearing at any of the FC outlets. $\rho$ is the fluid density. The momentum considered was the one acting on the vertical direction. The net momentum acting on the jet entering the mixing chamber (MC) is obtained when considering the vertical forces defined by Equation (14) acting instantaneously on both FCs' outlets. In this study, the fluid is considered as incompressible, and fluid density is time- and spatial-independent, but fluid velocity and pressure are spatial- and temporal-dependent. Therefore, to maximize the precision of the calculations, the instantaneous momentum equation to be applied at each FC outlet was discretized and evaluated at each grid cell belonging to the corresponding surface. The resulting equation reads as the following:

$$F = F_{massflow} + F_{pressure} = \rho \sum_{i=1}^{i=n} (S_i V_i^2) + \sum_{i=1}^{i=n} P_i S_i \tag{15}$$

The subindex $i$ denotes any mesh cell belonging to the FC outlet surface. The term $n$ defines the total number of cells corresponding to any of the two FC outlet surfaces. $P_i$ and $V_i$ characterize the instantaneous pressure and velocity components acting on the corresponding mesh cell in vertical direction. In the present paper, the instantaneous momentum term due to the mass flow was obtained simply by adding the elementary momentum terms of each mesh cell belonging to the chosen surface. The momentum pressure term was obtained when multiplying the instantaneous static pressure acting on each cell by the cell surface and then adding the elemental momentum pressure terms

corresponding to the surface under study. Only the momentum in the vertical direction was considered in all cases.

## 6. Results

### 6.1. Reynolds Number Modification

One of the main characteristics of any fluidic oscillator is its linear relation between the inlet Reynolds number and the FO output mass flow frequency. Four Reynolds numbers, Re = 27,483, 41,224, 54,595, and 68,707, based on the mixing chamber inlet width and velocity, were simulated to evaluate the actual FO linear performance. The result was a perfect linear behavior defined by the equation $f(Hz) = 0.0005364786 \times Re - 0.18617977$ with a correlation coefficient of $R^2 = 0.999966$. It needs to be considered that this equation is valid in the range of Reynolds numbers presented here.

The unsteady (time-dependent) mass flow at the FO outlet and at the lower FC channel outlet are introduced in Figure 7 for the different Reynolds numbers studied. As the Reynolds number increases, the FO outlet mass flow average value and peak-to-peak amplitude keeps increasing (Figure 7a) amplitude increases by 270% when the Reynolds number raises from 27,483 to 68,707. The peak-to-peak amplitude of the FC mass flow suffers a much smaller growth with the Reynolds number increase: only 159% when comparing the minimum and maximum Reynolds numbers studied, with an average value around zero. In fact, the mass flow across the FC is non-symmetric: the mass flow flowing towards the FC outlet at Reynolds number 68,707 is about 36% larger than the one flowing towards the FC inlet, and the percentage raises to almost 68% for the minimum Reynolds number studied (see Figure 7b). To unveil why the FO and FC mass flow amplitudes and frequencies change at different Reynolds numbers, we need to understand the origin of the self- sustained oscillations. In this regard, we will next focus on studding the stagnation pressure fluctuations inside the MC. Perhaps one of the most relevant parameters to consider is the stagnation pressure unsteadiness at the mixing chamber (MC) outlet's inclined walls. The stagnation pressure oscillations appear to be directly correlated with the momentum pressure and mass flow terms measured at the FC outlet. These three parameters are presented in Figure 8a,c,e, respectively. On the right-hand side of Figure 8, the time-averaged values of the MC lower inclined wall stagnation pressure, the net momentum time-averaged pressure term measured at the lower FC outlet, and the time-averaged mass flow momentum measured at the same location are presented (see Figure 8b,d,f, respectively). These figures also show the peak-to-peak amplitude of these variables as a function of the Reynolds number evaluated. The unsteady net momentum obtained when considering the pressure and mass flow terms measured simultaneously at both feedback channel outlets and for all Reynolds numbers evaluated is presented in Figure 8g, the corresponding time-averaged and peak-to-peak amplitude values are introduced in Figure 8h.

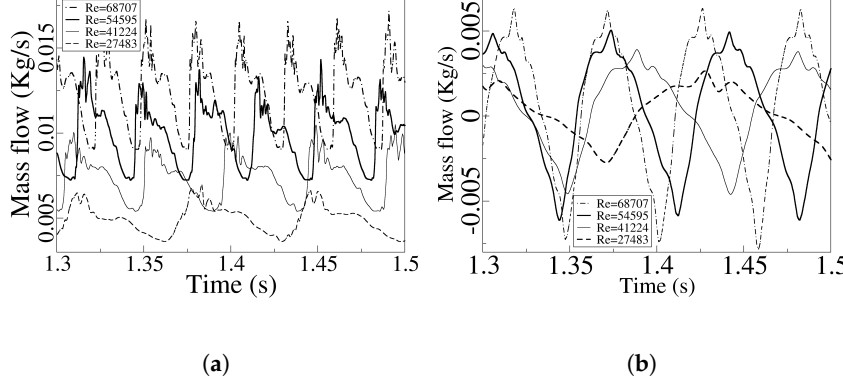

(a)  (b)

**Figure 7.** Fluidic oscillator outlet unsteady mass flow (**a**) and feedback channel mass flow (**b**) as a function of the Reynolds number.

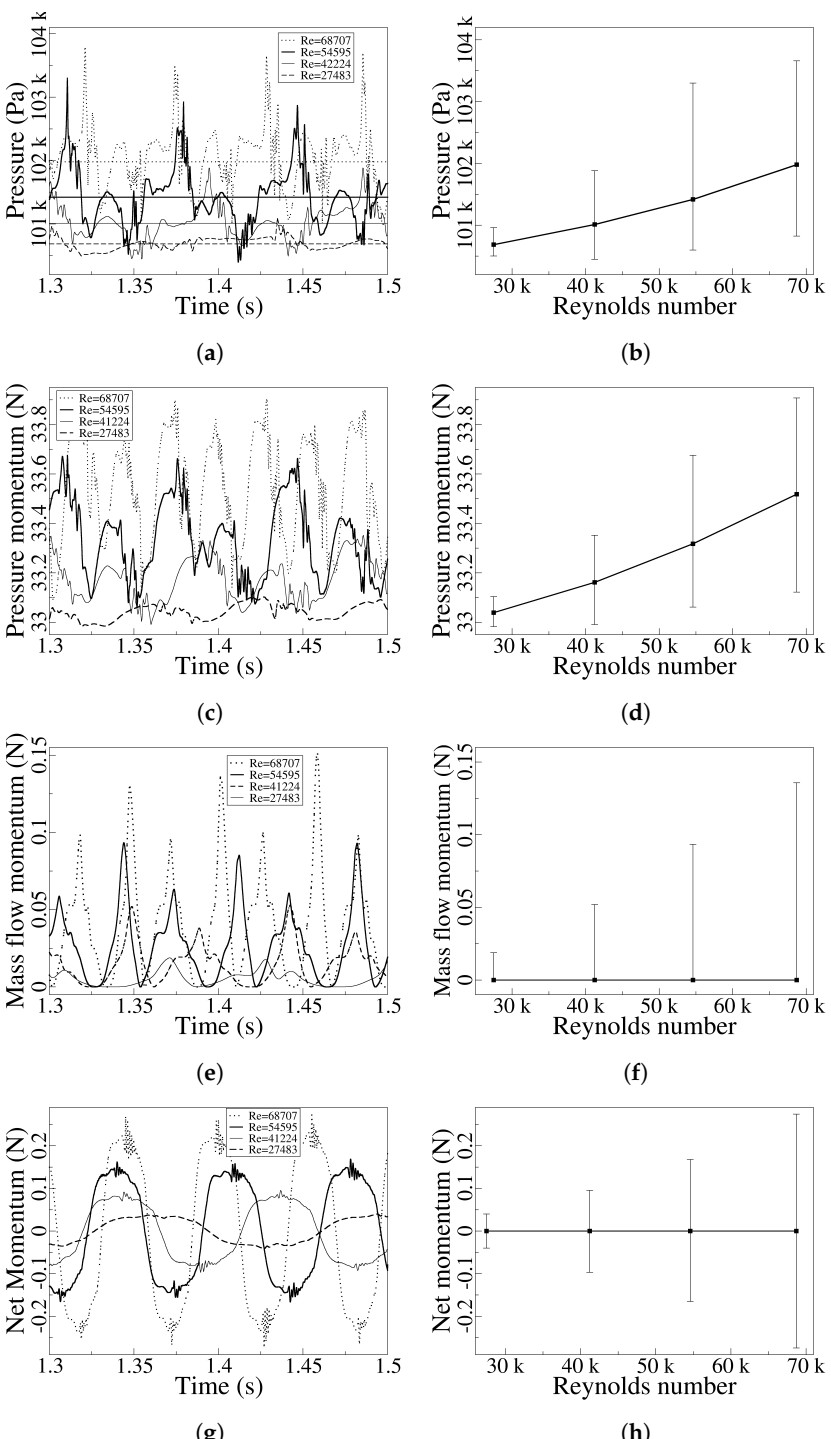

**Figure 8.** Unsteady stagnation pressure measured at the mixing chamber lower inclined wall (**a**), time-averaged values and peak-to-peak amplitudes (**b**). Pressure momentum terms measured at the FC lower outlet (**c**) and their respective average values and peak-to-peak amplitudes (**d**). Dynamic mass flow momentum terms measured at the FC lower outlet (**e**), and their respective average values and peak-to-peak amplitudes (**f**). Instantaneous net momentum obtained when considering pressure and mass flow momentum terms at both FC outlets (**g**), average values, and the peak-to-peak amplitudes (**h**). All graphs consider the four Reynolds numbers studied.

From the unsteady graphs, it appears as if the frequencies associated with the stagnation pressure term measured at the MC inclined wall and for the different Reynolds numbers are also observed in the unsteady values representing the pressure momentum,

mass flow momentum, and net momentum values shown in figures Figure 8c,e,g, respectively, showing a correlation between all these parameters.

In fact, the frequencies associated with the different parameters and different Reynolds numbers are presented in Table 5, where it is demonstrated that a correlation between all these parameters has to exist. Notice that the main frequency is almost identical for all the parameters evaluated.

**Table 5.** Main frequency as a function of the Reynolds number and the unsteady parameter evaluated.

| Reynolds Number | Re = 27,483 | Re = 41,224 | Re = 54,595 | Re = 68,707 |
|---|---|---|---|---|
| **Parameter** | **Main Frequency [Hz]** | | | |
| MC wall pressure | 7.5 | 11 | 14.53 | 18.55 |
| FC Pressure momentum | 7.52 | 10.86 | 14.62 | 18.57 |
| FC Mass flow momentum | 7.52 | 10.88 | 14.53 | 18.26 |
| Net momentum | 7.25 | 10.85 | 14.6 | 18.25 |

To further evaluate the link between the four parameters presented in Figure 8, the reader should now focus on Figure 8b,d,f. When comparing the peak-to-peak amplitude for the three parameters introduced in these figures and considering the minimum and maximum Reynolds numbers analyzed, it is observed that the maximum peak-to-peak amplitude is about 6.5 times larger than the minimum one, which corresponds to the minimum Reynolds number. In other words, the mixing chamber inclined wall stagnation pressure unsteadiness appears to be driving the momentum pressure term unsteadiness as well as the mass flow momentum term unsteadiness measured at the FC lower outlet. At this point and based on the information presented in Table 5 and Figure 8, we can conclude that the self-sustained oscillations appear to be controlled by the stagnation pressure unsteadiness at the MC inclined walls. To further reinforce this hypothesis, it is relevant to highlight that the momentum pressure term measured at a FC outlet width is nearly three orders of magnitude higher than the mass flow one Figure 8c,e. The net momentum is very small compared with the pressure momentum term measured on a FC single outlet, but it is about four times larger than the mass flow momentum of a single FC outlet, compare Figure 8g,e. This clarifies that small variations in the pressure momentum have a deep effect on the net one. To understand the relevance of the pressure net momentum terms versus the mass flow ones measured instantaneously at both FC outlets, Figures 9 and 10 were generated.

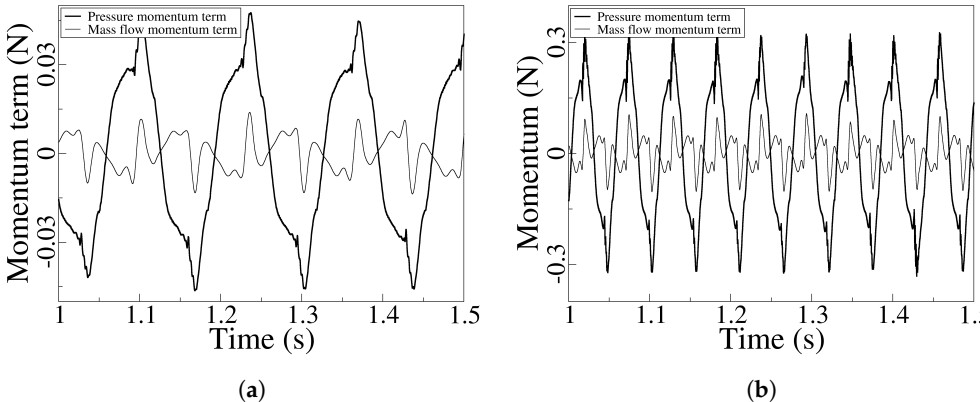

(a)                                                            (b)

**Figure 9.** Instantaneous net momentum pressure and mass flow terms measured at the feedback channels outlet sections and for the minimum and maximum Reynolds numbers studied. (**a**) Reynolds number 27,483. (**b**) Reynolds number 68,707.

The first of the two figures compares the two unsteady net momentum terms for the minimum and maximum Reynolds numbers studied, where it states that the value of the net

momentum pressure term at Re = 27,483 is 3.27 times larger than the net momentum mass flow term. In fact, the relation between the pressure and mass flow net momentum terms remains rather constant for all Reynolds numbers studied; this is clearly observed in Figure 10, where the peak-to-peak amplitude of both net momentum terms is presented for the four Reynolds number evaluated. All these graphs clarify that for the FO configuration presented in this study, the forces acting onto the jet as it enters the mixing chamber are mostly due to the pressure operating at the FC outlets, yet the forces due to the FC mass flow term are of the same order of magnitude as the pressure ones. At this point, it is important to recall the work performed by [25–27], where it was stated that the jet was pressure-driven and mass flow forces were playing a secondary role. The much more relevant role played by the FC mass flow forces in the present FO configuration is believed to be due to the particularly short and wide feedback channels employed in the present fluidic oscillator.

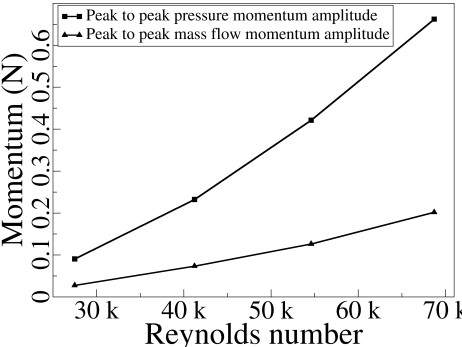

**Figure 10.** Pressure and mass flow peak-to-peak net momentum amplitude measured at the feedback channels for the four Reynolds numbers evaluated.

Instantaneous FO velocity and pressure fields for the minimum and maximum Reynolds numbers evaluated, Re = 27,483, and Re = 68,707, are presented in Figure 11. The instant chosen is the one at which the stagnation pressure value at the MC lower inclined wall is about the maximum (notice the red spot both pressure fields show at this location). It is also relevant to observe that both lower FCs are pressurized when compared with the respective upper ones. In fact, this figure clarifies that as the Reynolds number increases, the maximum and minimum pressure inside the mixing chamber increases and decreases, respectively (see the color bars associated), therefore explaining the peak-to-peak amplitude increase in all parameters introduced. Four videos showing the FO unsteady velocity and pressure fields for these two Reynolds numbers are introduced in the appendix as Supplementary Materials. The videos clarify the rather large reversed flow (from the FC outlet towards the FC inlet) previously introduced in Figure 7b; they also show the temporal evolution of pressure and mass flow in the FCs and the mixing chamber.

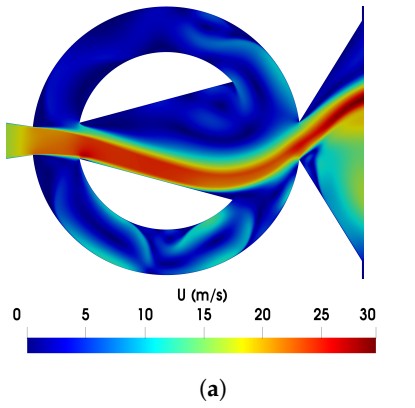
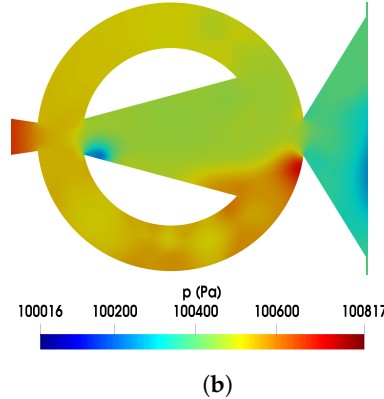

(a)                                                    (b)

**Figure 11.** *Cont.*

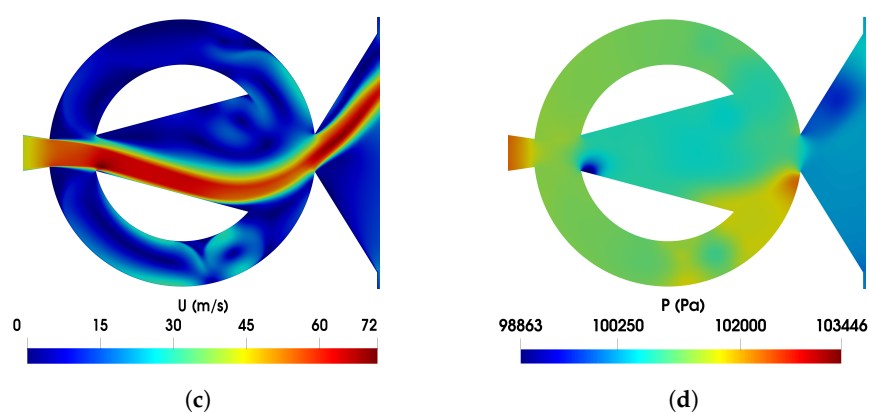

(**c**)  (**d**)

**Figure 11.** Fluidic oscillator internal velocity magnitude field (**a**,**c**) and pressure field (**b**,**d**). Minimum Reynolds number $Re = 27{,}483$ (**a**,**b**), and maximum Reynolds number $Re = 68{,}707$ (**c**,**d**).

*6.2. Outlet Width Modification*

In the present subsection, the same flow parameters previously evaluated as a function of the Reynolds number will now be analyzed for several outlet widths (six in total) initially defined in Table 3. Notice that in the present section the Reynolds number is kept constant at Re = 54,595. As previously performed for different Reynolds numbers, Figure 12 introduces the FO and FC unsteady mass flows for the smallest (outlet 1), largest (outlet 6), and standard (STD) outlet widths evaluated in the present study. As expected, the FO outlet mass flow average value remains constant, and the incoming flow has a constant Reynolds number. However, the peak-to-peak amplitude drastically grows with the outlet width; in fact, the FO outlet mass flow amplitude grows by 286% when comparing the largest (outlet 6) and smallest (outlet 1) widths evaluated (Figure 12a). It appears that the FO outlet mass flow amplitude is linked with the degree of freedom the jet has inside the MC. The FO oscillation frequency reduces by 10% with the outlet width increase; it went from 14.8 Hz in (outlet 1) to 13.33 Hz in (outlet 6). In fact, considering that the mass flow entering the FO remains constant, a reduction in frequency is to be expected when increasing the oscillation amplitude. On the other hand, the FC mass flow (see Figure 12b) appears to be rather unaffected by the outlet width modifications tested, the FC mass flow average value and peak-to-peak amplitude and frequency suffer negligible changes of less than 0.5% when comparing the maximum and minimum outlet widths studied. At this point, we conjecture that the rather constant FC mass flow, could be due to a constant pressure drop between the respective feedback channels inlet and outlet sections.

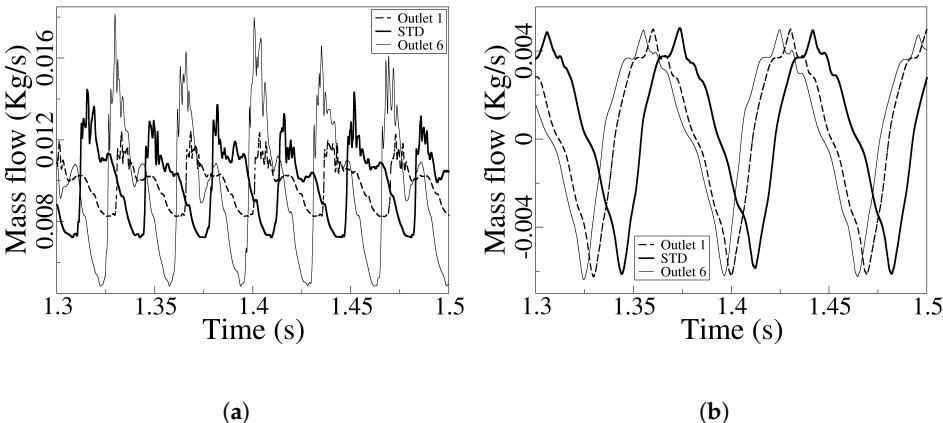

(**a**)  (**b**)

**Figure 12.** Fluidic oscillator outlet unsteady mass flow (**a**) and feedback channel mass flow (**b**) for the smallest (outlet 1), largest (outlet 6) and standard (STD) outlet widths evaluated.

Perhaps it is very relevant to highlight that the FC mass flow is not always flowing from the FC inlet towards the FC outlet, but there is an even larger instantaneous reverse flow

(flow going from the FC outlet towards the FC inlet). This large reverse flow is more clearly seen in the velocity field videos presented in the appendix as Supplementary Materials, and it seems to be a particular characteristic of the present FO configuration. Notice that the wide and short FC paths enhance the traveling of the fluid particles in both FC directions. On the other hand, the MC inlet width chosen for the present FO configuration appears to be rather small and so it favors the existence of reverse flow. It is also relevant to highlight the existence of flow asymmetry on both the FO outlet and FC mass flows. When the FO outlet mass flow reaches its maximum, the peak is rather scattered, as if the fluid was trembling, but when the FO outlet mass flow reaches its minimum, which happens when the FO outlet mass flow flows quite horizontally, the reverse flow in one of the FC and the direct flow in the other FC are both at the maximum. Under these conditions, the FO outlet mass flow presents rather a smooth minimum peak. In reality, the FO outlet mass flow trembling is generated when the reverse feedback channel mass flow reaches the FC inlet and interacts with the main jet flowing inside the MC. This interaction happens exactly a quarter of a cycle after the instant at which the reverse FC mass flow is at the maximum and corresponds to the time needed by the FC mass flow to travel from the FC inlet to the FC outlet. When these two jets flows interact, the stagnation pressure at the MC converging walls suffers a huge fluctuation which affects the entire domain, also generating the trembling on the FO outlet mass flow. These details can be seen in the videos given in the appendix as Supplementary Materials and will be carefully addressed in the final section of the paper.

The unsteady stagnation pressure measured at the MC outlet lower inclined wall is presented in Figure 13a, from where it can clearly be seen that as the outlet width is reduced, the time-averaged value and the peak-to-peak amplitude at the MC outlet inclined wall raise. In fact, the time-averaged pressure at the entire MC and FC's also increases. The time-averaged evolution of the unsteady pressure at the MC outlet inclined wall, as well as its peak-to-peak amplitude for all outlet widths studied is presented in Figure 13b. When comparing the minimum and maximum outlet widths, time-averaged and peak-to-peak pressure increases of almost 2.3% and 76%, respectively, are observed. A very similar trend is observed when analyzing the unsteady and time-averaged pressure momentum term at the lower FC outlet (see Figure 13c,d). Notice that the time-averaged and peak-to-peak values increase (when comparing the minimum and maximum outlet widths), is almost 2% and 74%, respectively, again suggesting a direct correlation between the stagnation pressure at the MC inclined walls and the one at the FC outlet. The mass flow momentum term measured at the lower FC outlet is introduced in Figure 13e. As already observed when evaluating the different Reynolds numbers, the mass flow momentum term is over two orders of magnitude smaller than the pressure momentum one (compare the figures in Figure 13e,c), therefore indicating the jet's high sensitivity to the pressure forces. However, when considering the instantaneous mass flow and pressure momentum terms at both feedback channels outlets, it is observed that the net momentum acting on the jet as it enters the mixing chamber is rather small (Figure 13f) its peak-to-peak value being about three times larger than the mass flow momentum one measured at the lower FC outlet. Notice as well that the net momentum is rather constant for the different outlet widths studied since the pressure drop between both FC outlets slightly changes with the outlet width modifications considered in the present study. At this point, it is relevant to highlight that if the net forces acting on the jet as it enters the MC suffer a minor variation for the different outlet widths studied (Figure 13f), but the FO outlet mass flow peak-to-peak amplitude drastically grows with the outlet width (Figure 12a). Such growth needs to be associated with the freedom the main jet has with the outlet width increase. In order to determine the main contributor of the jet self-sustained oscillations, the instantaneous pressure momentum term evaluated on both FC outlets is presented for the minimum (outlet 1) and maximum (outlet 6) outlet widths in Figure 14a,b, respectively. For comparison, the unsteady mass flow momentum measured at the same locations and for the same outlets is also presented in the same two figures.

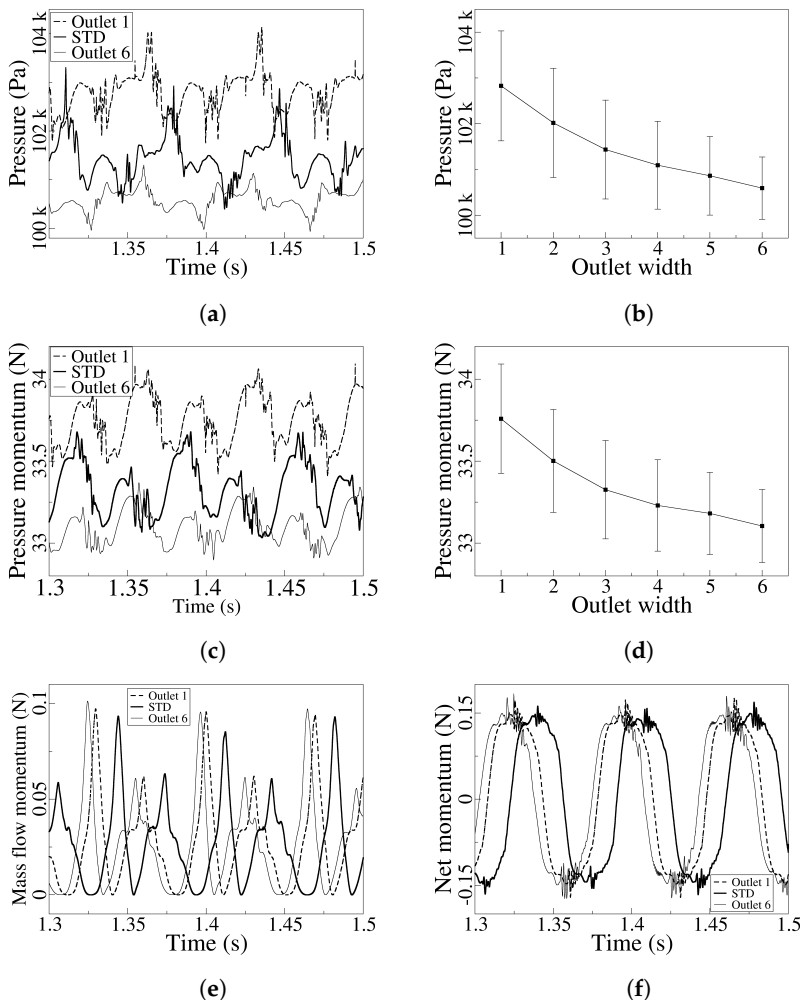

**Figure 13.** Unsteady stagnation pressure measured at the mixing chamber lower inclined wall (**a**), time-averaged values and peak-to-peak amplitudes (**b**). Pressure momentum terms measured at the FC lower outlet (**c**) and their respective average values and peak-to-peak amplitudes (**d**). Dynamic mass flow momentum terms measured at the FC lower outlet (**e**) and the instantaneous net momentum obtained when considering pressure and mass flow momentum terms at both FC outlets (**f**). All graphs consider three different outlet widths, the smallest (outlet 1), the standard case (STD) = (outlet 3), and the largest one (outlet 6) studied in the present study.

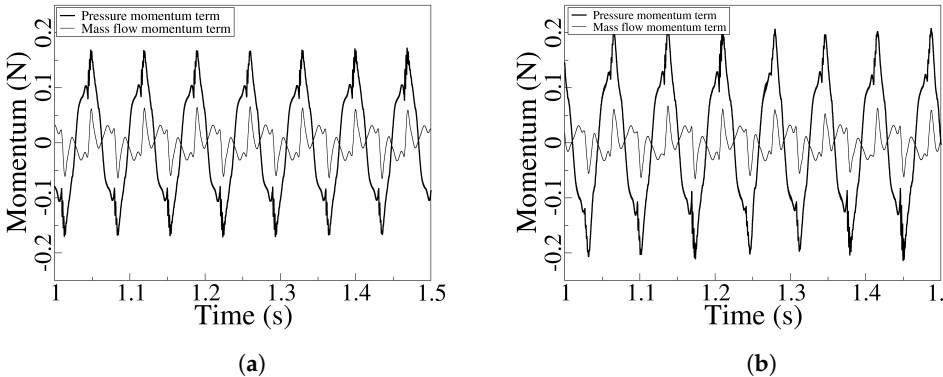

**Figure 14.** Instantaneous momentum pressure and mass flow terms measured at both feedback channels outlet sections and for the minimum and maximum outlet widths studied. (**a**) Minimum outlet width (outlet 1). (**b**) Maximum outlet width (outlet 6).

For the smallest outlet width considered (outlet 1), the peak-to-peak pressure momentum term is about 270% larger than the mass flow one. In fact, the pressure momentum term slightly grows with the outlet width while the mass flow term remains rather constant. Their relation is about 345% for the maximum outlet width (outlet 6). The main conclusion from Figure 14 is that the net forces acting on the jet as it enters the mixing chamber are due to the pressure, but the mass flow forces have the same order of magnitude. Instantaneous velocity and pressure fields for the minimum outlet width (outlet 1), standard outlet width, and maximum outlet width (outlet 6) are presented in Figure 15. The maximum velocity and the maximum pressure are observed to decrease with the outlet width increase. Particularly for the minimum outlet width (outlet 1), the entire FO is pressurized. Regardless of the outlet width, we can clearly see the high stagnation pressure area on the mixing chamber lower inclined wall, which in fact is what triggers the self-sustained oscillations.

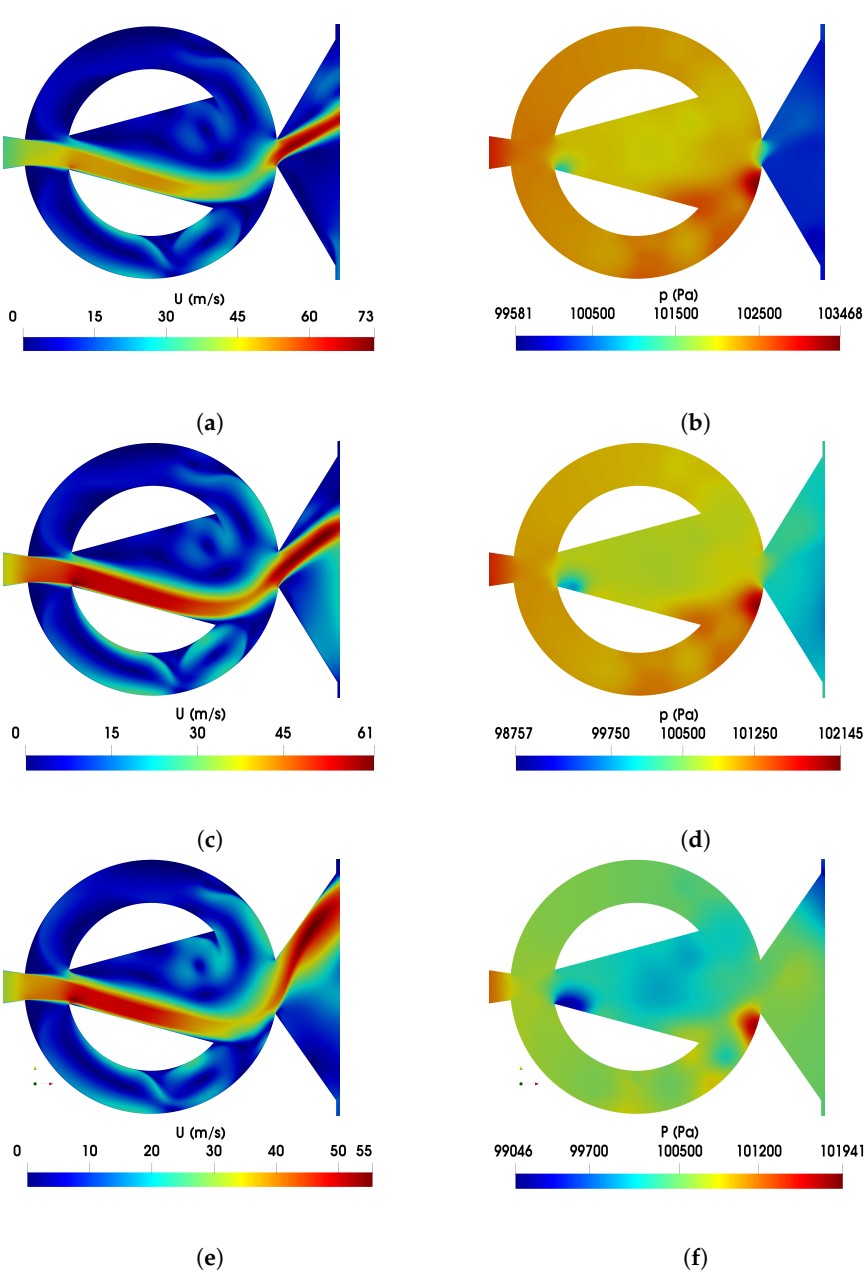

**Figure 15.** Fluidic oscillator velocity magnitude (**a**,**c**,**e**) and pressure fields (**b**,**d**,**f**), for the minimum (outlet 1) (**a**,**b**), standard (**c**,**d**), and maximum (outlet6) (**e**,**f**) outlet widths evaluated. Reynolds number is kept constant at Re = 54,595.

To more properly see the dynamics of the velocity and pressure fields, six videos which correspond to the three cases presented in Figure 15 are introduced in the appendix as Supplementary Materials. From the videos, we can clearly observe the reverse feedback channel mass flow appearing in all the cases studied. As this FO is a novel design that has never been studied at these Reynolds numbers, it clearly requires future work which could address the required dimensional modifications to reduce or even suppress the reverse flow in the feedback channels. In fact, the design presented here was mostly created to be able to evaluate the origin of the forces generating the self-sustained oscillations.

### 6.3. Mixing Chamber Internal Angle Modification

The present section clarifies the fluidic oscillator performance when the mixing chamber inlet angle is modified; the main details of the different MC angles studied were already specified in Table 4. As in the previous subsection, the Reynolds number will be kept constant at $Re = 54{,}595$. The fluidic oscillator outlet unsteady mass flow for the mixing chamber minimum and maximum angles (Angle 1) and (Angle 3), respectively, and for the standard one (STD), are presented in Figure 16a. The time dependent feedback channel mass flow measured at the lower feedback channel outlet and for the same three mixing chamber angles is introduced in Figure 16b. As the mixing chamber angle increases, the peak-to-peak amplitude of the FO outlet mass flow as well as its associated frequency keep increasing. An opposite effect is observed on the FC mass flow: since the peak-to-peak amplitude decreases with the MC angle increase—also decreasing the maximum reverse flow in the feedback channels—the frequency increases. The outlet mass flow peak-to-peak amplitude and frequency increase when comparing the minimum (Angle 1) and maximum (Angle 3) angles studied, which are 46% and 15%, respectively, and the corresponding FC mass flow amplitude decrease and frequency increase are 65% and 19.5% respectively. It is perhaps more interesting to highlight the fact that the maximum outlet mass flow appears when the jet leaving the mixing chamber is experiencing either its maximum or minimum inclination angle versus the horizontal axis; this is more clearly observed when looking at the videos (see appendix as Supplementary Materials).

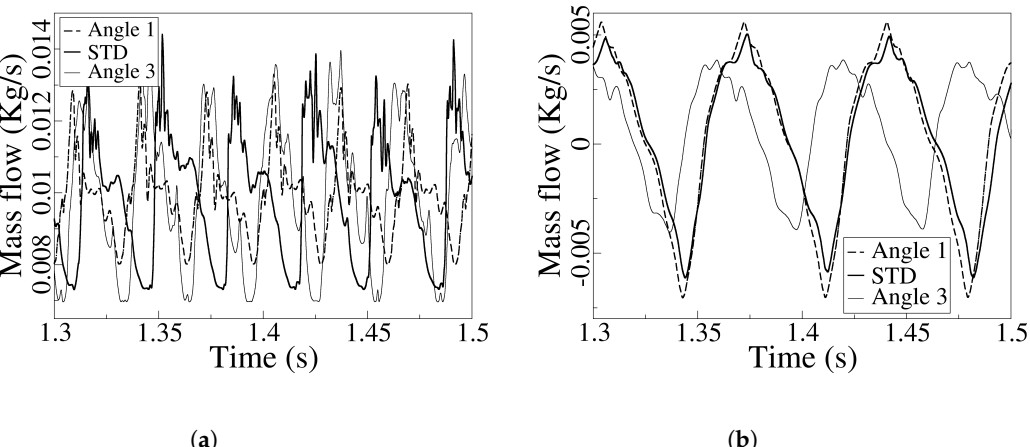

(**a**)

(**b**)

**Figure 16.** Fluidic oscillator outlet unsteady mass flow (**a**) and feedback channel mass flow (**b**), for the smallest mixing chamber internal angle (Angle 1), the largest MC internal angle (Angle 3), and the standard case (STD).

In fact, the same phenomenon was happening in Figure 7a for different Reynolds numbers and in Figure 12a for different outlet widths. Notice that for these two figures as well as for Figure 16a, the respective minimum mass flows obey to the instants the jet leaving the FO flows horizontally. Two maximum consecutive peaks characterize the instants at which the jet leaving the FO generate the maximum angles, whether positive or negative, versus the horizontal axis. This is why in all these graphs the FC mass flow seems as if the corresponding frequency was half of that of the FO outlet mass flow. A

final remarkable detail to be observed in Figure 16b is that the reverse flow (flowing from the FC outlet towards the FC inlet) keeps decreasing with the increase in the MC angle. It appears as if a higher freedom in the movement associated with the jet inside the mixing chamber reduces the reverse flow in the feedback channels, although in reality, this effect will be more clearly understood when analyzing the pressure at the mixing chamber outlet inclined walls.

At this point, it is relevant to compare the effects on the FC mass flow when the FO outlet width is increased and when the MC internal angle is increased (see Figure 12b and Figure 16b). When increasing the FO outlet width, it generates a negligible effect on the FC mass flow peak-to-peak amplitude, but this reduces when increasing the MC inlet angle. In both cases, the outlet mass flow amplitude increases, and in both cases, the jet inside the MC has a higher degree of movement. Yet, the increase in the MC internal angle affects both the FC mass flow and the FO mass flow amplitude while the increase in the FO outlet width just affects the FO outlet mass flow, probably due to the fact that the degree of freedom inside the mixing chamber remains quite unchanged.

For the same three MC angles already introduced in Figure 16, the unsteady maximum stagnation pressure measured at the MC outlet lower inclined wall, the pressure momentum term measured at the lower FC outlet, the mass flow momentum term measured at the same location, and the net momentum acting over the jet as it enters the mixing chamber (which considers the instantaneous pressure and mass flow terms at both FC outlets) are presented in Figure 17a–d, respectively.

Focusing on the stagnation pressure (Figure 17a), it is observed that as the MC inclined angle increases, the stagnation pressure peak-to-peak amplitude decreases, the maximum pressure becomes smaller, and the signal appears to have a higher degree of randomness. A similar trend is observed when evaluating the pressure momentum term at the FC lower outlet. The trend is also similar when considering the mass flow momentum at the FC lower outlet and the overall net momentum acting over the jet as it enters the MC, yet these two momentum signals are smoother than the previous pressure signals described since they are obtained from the integration of the respective values across the FC outlet surfaces. These effects seem understandable when considering that the increase in the MC internal angle brings a higher degree of freedom to the jet inside the MC and, as a result, the jet impinges in a wider area along the MC converging walls. The stagnation pressure has a higher associated randomness, the maximum stagnation pressure becomes smaller, and less particles impinge in a given point and at a given instant. The same pattern is observed in the rest of the variables since all of them correlate with the unsteady stagnation pressure.

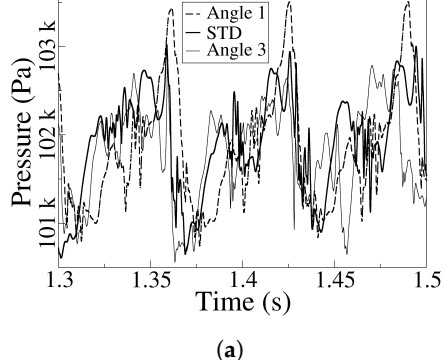

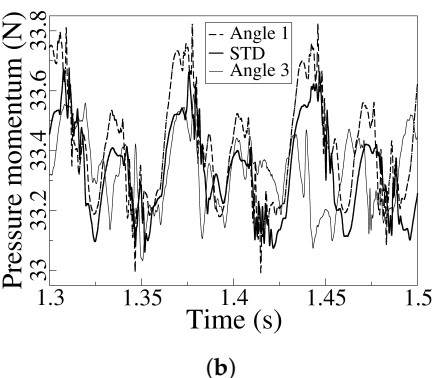

(a)                                                        (b)

**Figure 17.** *Cont.*

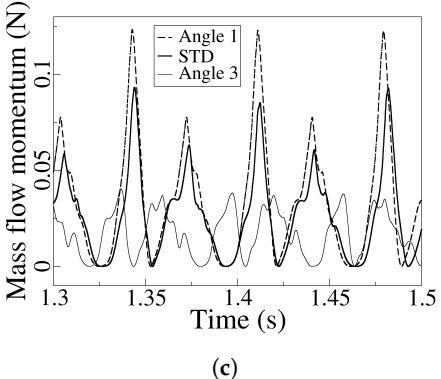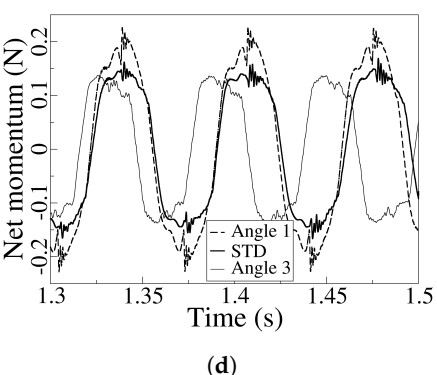

**(c)**　　　　　　　　　　　　　　　　　　**(d)**

**Figure 17.** Unsteady stagnation pressure measured at the mixing chamber lower inclined wall (**a**). Pressure momentum term measured at the FC lower outlet (**b**). Dynamic mass flow momentum term measured at the FC lower outlet (**c**) and the instantaneous net momentum obtained when considering pressure and mass flow momentum terms at both FC outlets (**d**). All graphs consider three different MC angles, the smallest (Angle 1), the standard case (STD) and the largest one (Angle 3) studied in the present study.

All graphs presented show a small increase in frequency as the MC inclined angle increases. When considering the FO outlet mass flow, the frequency increase is about 15%; such frequency increase rises to around 20% if the FC mass flow is evaluated. As already observed when analyzing the Reynolds and the MC outlet width cases, the pressure momentum term measured at the FC lower outlet (Figure 17b) is several orders of magnitude higher than the mass flow momentum term (Figure 17c) measured at the same location. Yet, when evaluating the mass flow momentum term measured instantaneously at both FC outlets and comparing it with the pressure momentum term analyzed in the same location (see the figures in Figure 18a,b, respectively) it can be concluded that both momentum terms have the same order of magnitude. The momentum pressure term is about 400% higher than the mass flow term. The corresponding velocity and pressure fields for the minimum (Angle 1) and maximum (Angle 3) angles evaluated are introduced in Figure 19. Small MC inclined angles direct the flow towards the MC outlet; this results in a higher stagnation pressure at the MC inclined wall (see Figures 17a and 19a,b). As the MC inclined angle increases, the jet inside the MC has a higher degree of freedom, this results in a higher degree of random oscillations and a smaller stagnation pressure at the MC inclined walls (see Figures 17a and 19c,d). Small MC inclined angles generate higher maximum instantaneous FO outlet velocities, but appearing only at the center of the jet, which entails smaller outlet mass flow amplitudes. Large MC inclined angles create outlet jets with much more uniform velocity distributions, being the FO outlet peak-to-peak amplitude larger, as observed when comparing Figures 16a and 19a,c.

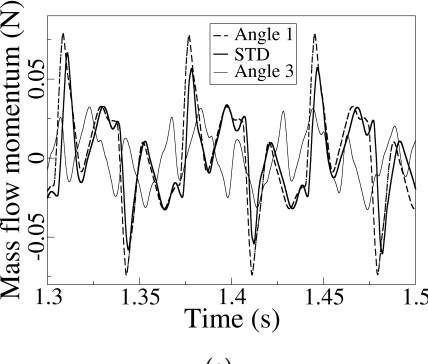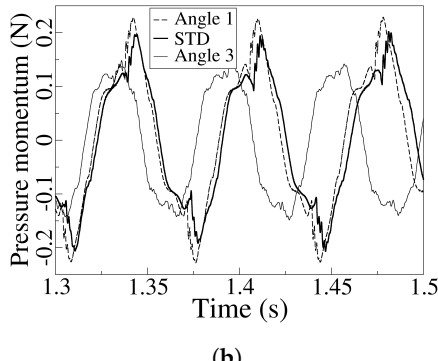

**(a)**　　　　　　　　　　　　　　　　　　**(b)**

**Figure 18.** Instantaneous mass flow and pressure forces measured at both feedback channels outlet sections and for the smallest MC angle (Angle 1), the standard case (STD), and the largest MC angle (Angle 3) studied. (**a**) Mass flow momentum. (**b**) Pressure momentum.

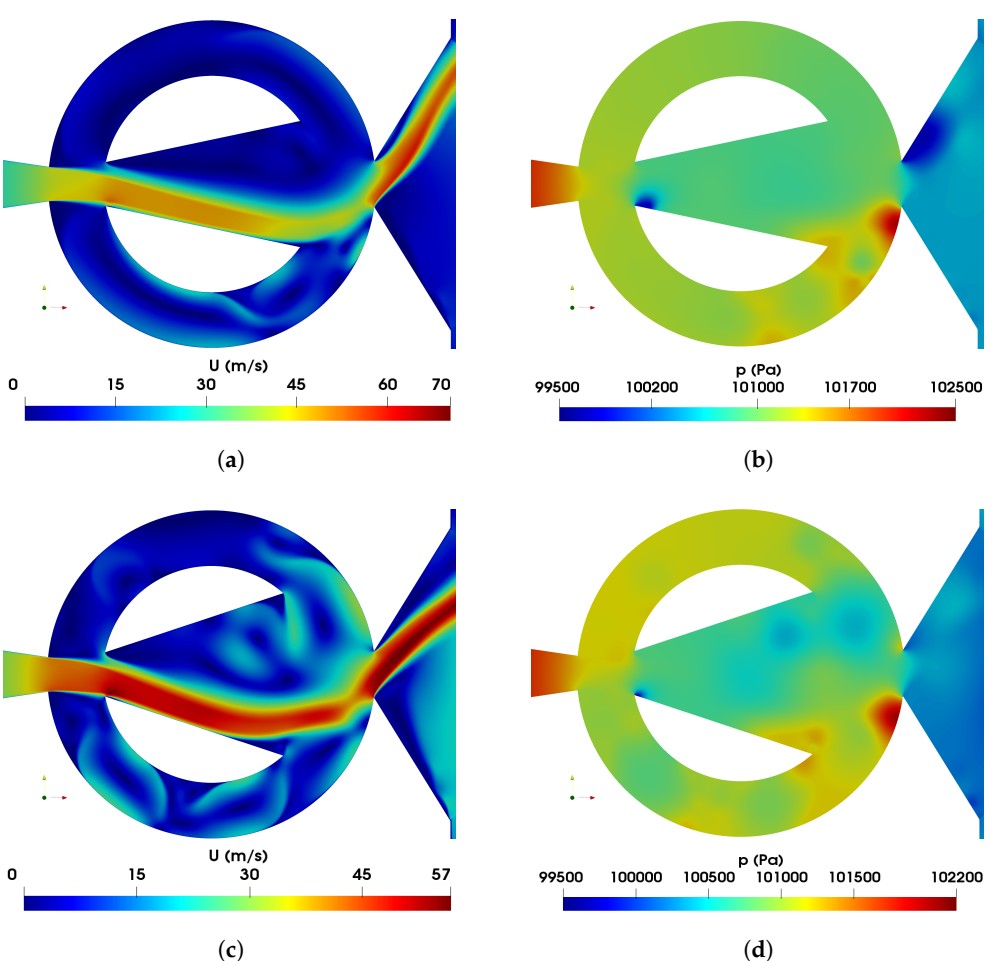

**Figure 19.** Fluidic oscillator internal velocity magnitude field (**a**,**c**) and pressure field (**b**,**d**). Minimum MC inclined angle (Angle 1) (**a**,**b**), Maximum MC inclined angle (Angle 3) (**c**,**d**).

## 7. Discussion of the Results, Origin of the Self-Sustained Oscillations

Until this point, the main unsteady characteristics of the FO have been analyzed at different Reynolds numbers, outlet widths, and MC inclined angles. For all cases studied, a large reverse flow at the FC has been observed (see Figures 7b, 12b and 16b). In fact, this large reverse flow was not previously observed in the FO configurations studied by [14,25–27] or even by [28], where the same FO configuration was studied but at a very low Reynolds number. In the present subsection, an oscillation cycle is carefully analyzed, linking the velocity and pressure fields at different instants with the forces acting over the jet; the concept here is to unveil the forces acting on the jet as it enters the mixing chamber and understand why the jet keeps oscillating, even when this large reverse flow exists. Figure 20 introduces the mass flow and pressure forces pushing the jet on each feedback channel outlet for an oscillation cycle in separate graphs, and the overall forces acting over the jet are also presented. Four different time instants, shown as vertical lines in these graphs, will serve to analyze the unsteady forces on the jet as well as the velocity and pressure fields associated, these fields are introduced in Figure 21. The oscillation period starts at time 1.375 s, the jet inside the MC is around its upmost position, the lower FC is still pressurized (Figure 21a,b) and the forces acting on the jet as it enters the MC are still pushing the jet upwards (Figure 20). In this figure, for convenience both FC pressure forces are represented as positive.

However, this situation is about to change since the upper FC reverse mass flow (which is almost at its maximum value) is about to interact with the jet inside the MC. Such interaction generates the pressure fluctuations observed in the pressure momentum graph shown in Figure 20 and are due to the temporal modification of the location and intensity of

the stagnation pressure point at the MC converging wall. Notice as well that at this instant the lower FC mass flow is around its maximum (Figures 12b and 16b). It is also relevant to note the large vortical structure appearing at the center of the MC, which appears to be pulling down the right-hand side of the jet inside the MC. The next time period presented, corresponds to time 1.390 s and 1/4 of the oscillation period. At this instant, the upper FC is pressurizing, the forces on the jet change from positive to negative as it enters the MC, the jet inside the MC is moving downwards, and the jet leaving the FO is around its lower angular position (Figure 21c,d). The FO outlet mass flow is at the maximum (Figure 12a), and the FC mass flow on both FCs is about zero (Figures 12b and 16b). On the next quarter of a period $T = 1/2$, time 1.41 s, the jet inside the MC is about to reach its lower position, the upper FC is pressurized (but this situation is about to change), then the reverse mass flow at the lower FC is going to interact with the main jet at the MC, as it happened at $T = 0$. Such jet interaction will generate pressure fluctuations which will finally change the direction of the forces acting on the jet entering the MC (see Figures 20 and 21e,f). The outcome of these effects is observed in Figure 21g,h (time 1.425 s, $T = 3/4$), in which it is observed that the lower FC is pressurizing. Both FCs' mass flows are about zero, the jet inside the MC is moving upwards, the jet at the FO outlet is about to reach its upper angular position, and the FO outlet mass flow is around its maximum value (Figure 12a).

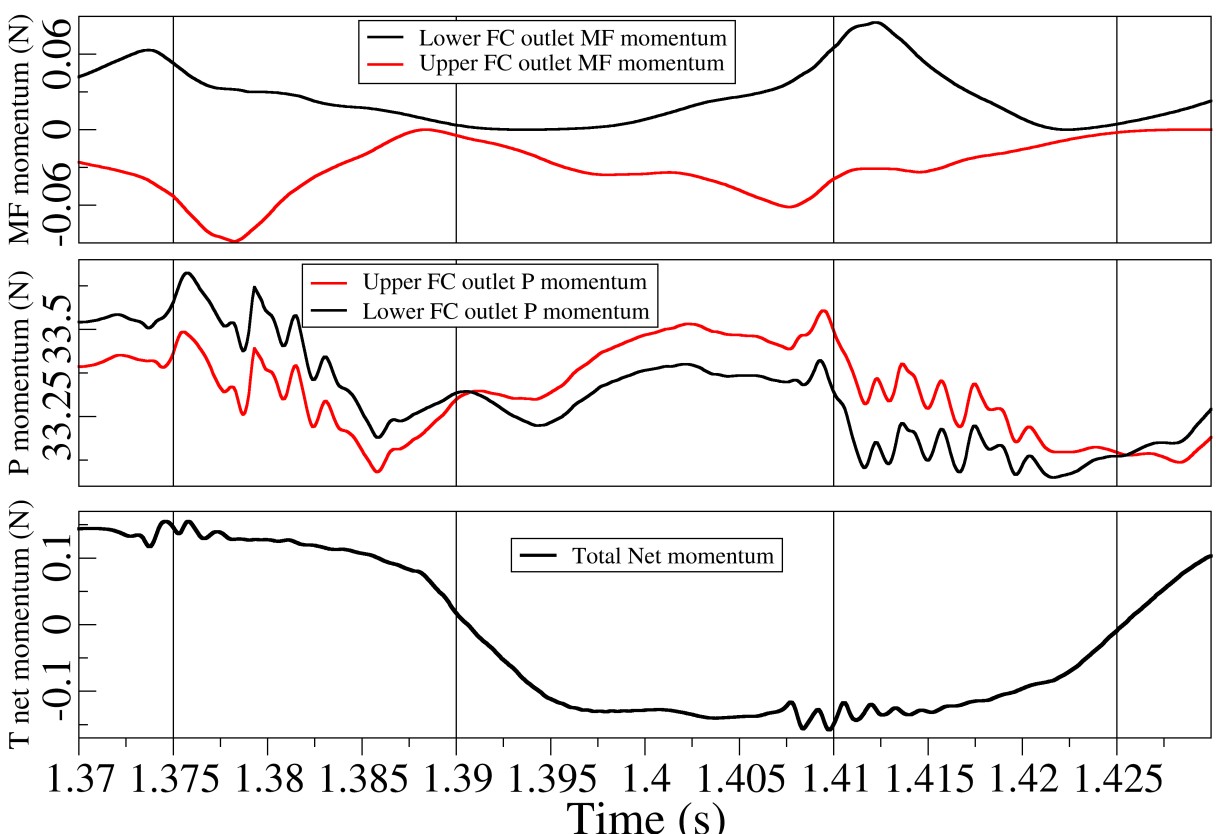

**Figure 20.** Unsteady mass flow and pressure forces measured at both feedback channel outlets and the overall forces acting on the jet; about half of a cycle is presented here. Baseline case configuration (STD) at $Re = 54,595$.

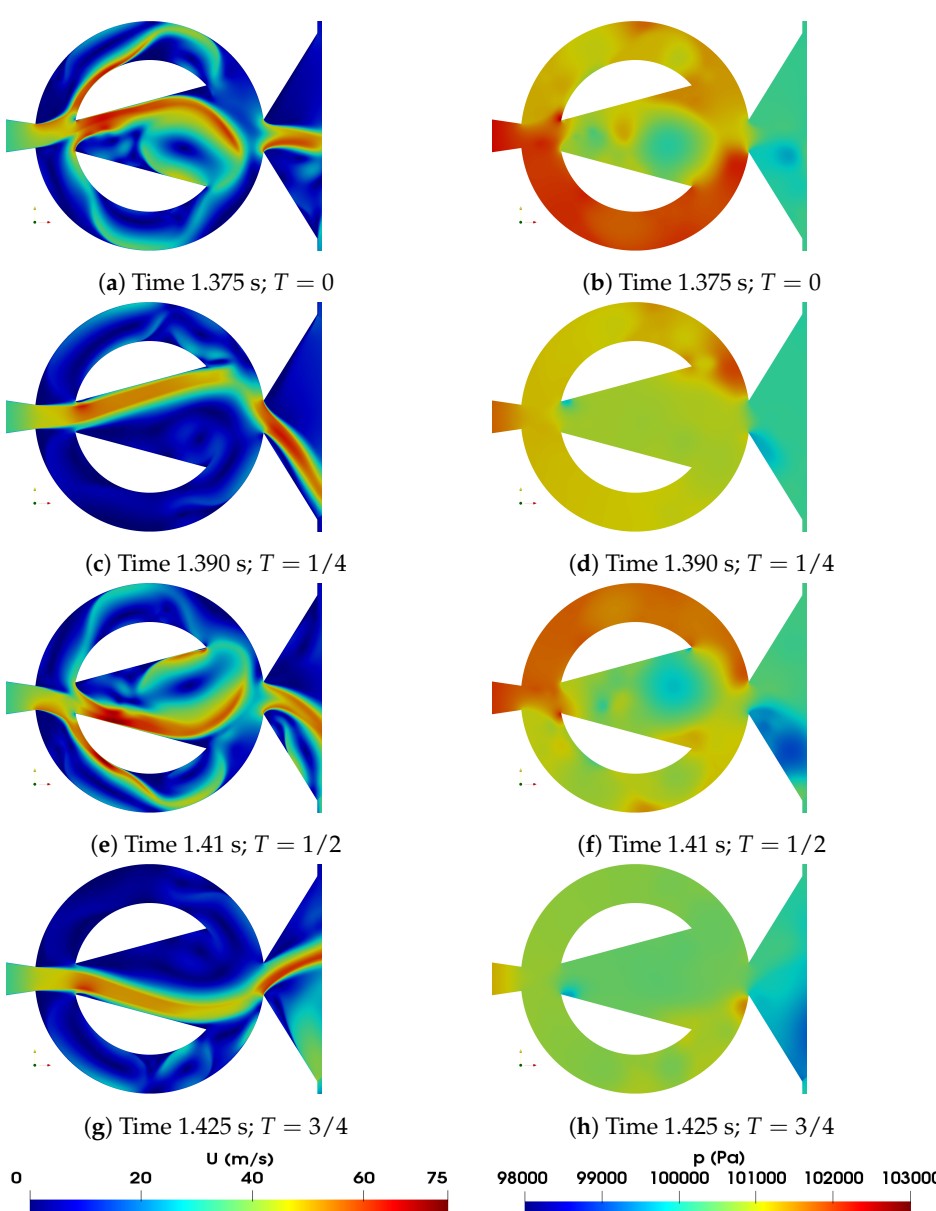

**Figure 21.** Velocity and pressure fields at different timesteps along a half oscillation period. Baseline case (STD) at Re = 54,595.

## 8. Conclusions

This paper presents a rather new configuration of a fluidic oscillator characterized by its simplicity and its wide and short feedback channels. Analysis has been performed for different incoming flow Reynolds numbers, different outlet widths, and several mixing chamber inclined angles. Despite its simplicity, the FO presents a very good linearity between the output flow frequency and the incoming Reynolds number. Regardless of the case studied, the onset of the self-sustained oscillations is proven to be mostly due to the pressure forces acting onto the jet as it enters the mixing chamber; the forces due to the mass flow, although smaller, have the same order of magnitude as the pressure forces in the present configuration. This is at odds with what was previously observed in [25–27], where the self-sustained oscillations were pressure-driven and the mass flow forces had a minor influence, but the FO studied in these references was a completely different one, with much longer and narrower feedback channels than the ones studied here. Another characteristic which differentiates the present configuration from the previously studied ones is the large reverse flow existing in the feedback channels; we have proven that this

reverse flow interacts with the main jet inside the mixing chamber, generating random pressure fluctuations in the mixing chamber. This happens just before the jet flips towards the opposite direction. The Reynolds number increases and generates increases in the outlet mass flow and feedback channel mass flow amplitude, and the pressure inside the mixing chamber is also rising. The forces acting on the jet also rise with the Reynolds number. The outlet width increase is associated with an increase in the outlet mass flow amplitude, yet the feedback channel mass flow suffers minor modifications. The stagnation pressure inside the MC as well as its peak-to-peak amplitude keep decreasing with the increase in the outlet width. The same trend is observed in the pressure forces acting on a FC outlet. When considering the modification of the mixing chamber internal angle, it is observed that larger MC internal angles are associated with larger outlet mass flow peak-to-peak amplitudes and smaller amplitudes of the feedback channel mass flow, also reducing the reverse flow in the FC. Pressure forces and mass flow forces acting on the jet as it enters the MC are also reduced with an increase the MC internal angle.

**Supplementary Materials:** The following supporting information can be downloaded at: http://www.youtube.com/@kavooskarimzadegan (accessed on 15 February 2024). A set of four videos introducing the flow and pressure fields initially presented in Figure 11 at Reynolds numbers 27,483 and 68,707 are introduced here. A second set of six velocity and pressure field videos presented here correspond to the standard, maximum, and minimum outlet widths for a constant Reynolds number of 54,595. Figure 15 introduced instant fields for these three cases. The final four videos given as Supplementary Material characterize the flow and pressure fields inside the FO for the minimum and maximum MC inclined angles studied. Figure 19 introduced these velocity fields. It is important to note that for all the cases studied, a large reverse flow is observed inside the FCs.

**Author Contributions:** Conceptualization, J.M.B.; methodology, K.K. and J.M.B.; software, K.K.; validation, K.K. and J.M.B.; formal analysis, K.K. and J.M.B.; investigation, K.K. and J.M.B.; data curation, K.K.; writing—original draft preparation, J.M.B., M.M. and K.K.; writing—review and editing, J.M.B.; visualization, K.K. and J.M.B.; supervision, J.M.B. and M.M.; project administration, J.M.B. All authors have read and agreed to the published version of the manuscript.

**Funding:** This research received no external funding.

**Institutional Review Board Statement:** Not applicable.

**Informed Consent Statement:** Not applicable.

**Data Availability Statement:** Data are contained within the article.

**Conflicts of Interest:** The authors declare no conflicts of interest.

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
