# Peer review of "Analysis of a Novel Fluidic Oscillator under Several Dimensional Modifications"

_applsci, doi:10.3390/app14051690_

Round 1

Reviewer 1 Report

Comments and Suggestions for Authors

Major comments:

1. The main concern about this problem setup is thet 2D modellng. In2D simulations, the fluctuations are artificially stong. The main point of the paper is how much oscilation is achived by the oscilator design, however, I do not thing a 2D simulation will be any acurate. Authors must convince a reader that A) fluctuations casued by the geometry are much stronger (by showing percentage) than 3D or B) compare with a 3D design or experiment to show the validity of their claims.

2. The applicability of the 2D geometry to a realistic design is not clear. What real application or design modification does this shape apply to. If built, would it be cylindrical or spherical representation? This must be clarified.

3. No comparison of the numerical results aree provided with experiment. I understand there might not exist experimental results for the oscilator design. But how about comparison of the numerical model with a baseline jet that would validate the model.

4. The authors provide inflow turbulance in the initial jet with inlet boundary condition prescribing nut of about 1e-3 m^2/s^2. There is no evidance or explanation provided on this, why these values are chosen? in reference to what? 

5. The authors must provide detailed evidance if inflow jet turbulance (artificially prescribed by the boundary condition) is effective on the oscilation achieved at the outet? If they change inflow BC how does the outflow fluctuations change? If not, show evidance or cite previous work.

6. Authors have not provided any clear information on the dimenion of the main jet nozzle, and how do the frequencies correlate to the actual dimensions of the jet. All results are shown in dimensional variables, without providing dimensions of the actual geometry. I suggest that all results shown in a non-dimensional normalized form. otherwise the results would lack scientific importancy and applicibility in a general sense.

Minor comments:

1. Fig. 2  is does not provide any useful nformation on grid sizing. if this is to show geometry, put dimensions. if it os to show mesh, show mesh in good quality.

2. I am sure authors meant "nozzle" NOT "nuzzle" in Fig 1

3. The figure after figure 15 does not have any titles. must be corrected

4. I suggest that time snapshots shown in Fig. 18 to be correlated to frequency (or non-dimensional reduced frequency) to provide any scientific importance. otherwise this is meaningless 

Comments on the Quality of English Language

There exist a few gramatical errors throughout the parer that are mentioned as minor comments above.

Author Response

Please see the document attached

Reviewer 2 Report

Comments and Suggestions for Authors

The present manuscript provides CFD results of the unsteady flow in a fluidic oscillator. The authors have well organized the present work, and the intro seems informative and comprehensive as well. The authors are advised to modify and improve the present work for pubilcation in journal.

1) The title of paper does not match with the present work. Readers can not understand the present work. There are no data on the performance of oscillator. 

2)  More detailed boundary conditions should be given and very careful validation must be made. Solution convergence should be clearly addressed and the time step applied should also be discussed in detail.

3) The authors should not try to simply list up the results obtained, but to discuss the underlying oscillation physics. 

4) Eq.(14) can be given as the force equation. Some terms are used inappropriately : dynamic mass flow, stagnation pressure dynamics, mass flow momentum, momentum dynamics,  two dynamic net momentum terms, pressure dynamics, dynamic stagnation pressure, dynamic mass flow momentum, etc, all of which is not easy to understand in terms of fluid dynamics.

5) The impact of the modification of oscillator shape should be analyzed in detail, in the point of view of fluid dynamics. 

Comments on the Quality of English Language

The present manuscript provides CFD results of the unsteady flow in a fluidic oscillator. The authors have well organized the present work, and the intro seems informative and comprehensive as well. The authors are advised to modify and improve the present work for pubilcation in journal.

1) The title of paper does not match with the present work. Readers can not understand the present work. There are no data on the performance of oscillator. 

2)  More detailed boundary conditions should be given and very careful validation must be made. Solution convergence should be clearly addressed and the time step applied should also be discussed in detail.

3) The authors should not try to simply list up the results obtained, but to discuss the underlying oscillation physics. 

4) Eq.(14) can be given as the force equation. Some terms are used inappropriately : dynamic mass flow, stagnation pressure dynamics, mass flow momentum, momentum dynamics,  two dynamic net momentum terms, pressure dynamics, dynamic stagnation pressure, dynamic mass flow momentum, etc, all of which is not easy to understand in terms of fluid dynamics.

5) The impact of the modification of oscillator shape should be analyzed in detail, in the point of view of fluid dynamics. 

Round 2

Reviewer 1 Report

Comments and Suggestions for Authors

Major comments: 

 The authors mentioned in the responce letter: "Although not presented in this paper, we performed a 3D simulation of the fluidic oscillator using around 7.7 million cells. We compared for the 2D and 3D simulations at Re = 54595, ...". If this is not published data must be presented here and validated. If published must be cited properly. Otherwise it does not hold scientific validity.

2. Also they mention: " We prefer not to include these graphs in the actual paper, simply because we want to generate a second document where, among other information, the 3D versus the 2D characteristics will be discussed in more detail. " I disagree with this approach for journal publications. I think they can present this work in a confrence paper, but if they wish to use this as evidence. It must be either reported here and validity of it being tested and approved otherwise if it is done somewhere else, it can be cited properly when that work is published. In other part of respone letter (page 5), it is mentioned " In fact the comparison between the 2D and 3D models, among other new findings is what we are planing to present in a next paper which is under development". My judgement of the manuscript can be solely based on the current mauscript not based on planned/under developement manuscript

3. The acclaimed "3D cases" can not be refered to as a solid reference by itself either. The authors are still using "  0.01 m" thickness and switching Boundary Conditions from "empty" in the OpenFoam bc definition, so this is just another method of performing 2D analysis that is previously attempted in various by many authors. The physical thickness of the domain is much smuller than wavelength of fluctuations. So any physical attempt of flow variation in 3rd dimension is automatically dampended by the geometry/boundary in 3rd dimension.

4. Major concern 4 is answered. but it should be mentioned in the manuscript.

5. Major concern 5. The produced figures in the responce letter must be present in the manuscript

6. Major concern 6 is NOT addressed in the revised manuscript.

Author Response

Please see the document attached

Reviewer 2 Report

Comments and Suggestions for Authors

From the first round reivew, the authors have replied to reviewer's comments and pointings out. Some of the authors' responses are acceptable but some are not reasonable. Thus, the authors should carefully reflect reviewer's comments in revising the manuscript. Otherwise, I can not recommend the present work for publication in journal. 

1) Eq.(1) ~(2)  are the incompressible version of NS, but Eq.(4)~(6) are the compressible one. Which version of NS have you used for the simulations?

2) M on he left side of Eq.(14) and (15) can be given by force F. This makes better sense to readers. Accordingly all the figures should be modified to this change.

3) There are no validation data, which can make the present data unreliable. 

4) The authors must well distinguish between "unsteady" and "dynamic". For ex.  what does the dynamic staganation pressure mean? The term of staganation stands for "static". Thus this term must be chaged to unsteady staganation pressure or fluctuating staganation pressure, etc. Otherwise, readers can be confusing with this kind of unusual terms. 

Comments on the Quality of English Language

From the first round reivew, the authors have replied to reviewer's comments and pointings out. Some of the authors' responses are acceptable but some are not reasonable. Thus, the authors should carefully reflect reviewer's comments in revising the manuscript. Otherwise, I can not recommend the present work for publication in journal. 

1) Eq.(1) ~(2)  are the incompressible version of NS, but Eq.(4)~(6) are the compressible one. Which version of NS have you used for the simulations?

2) M on he left side of Eq.(14) and (15) can be given by force F. This makes better sense to readers. Accordingly all the figures should be modified to this change.

3) There are no validation data, which can make the present data unreliable. 

4) The authors must well distinguish between "unsteady" and "dynamic". For ex.  what does the dynamic staganation pressure mean? The term of staganation stands for "static". Thus this term must be chaged to unsteady staganation pressure or fluctuating staganation pressure, etc. Otherwise, readers can be confusing with this kind of unusual terms. 

Author Response

Please see the document attached

Round 3

Reviewer 1 Report

Comments and Suggestions for Authors

After reviewing the authors responces and the inclusion of multiple revisoins, I recommend publication of the final version of the modified manuscipt.

Reviewer 2 Report

Comments and Suggestions for Authors

The authors have improved the present manuscript based on reviewer's comments. I recommend the present work for publication in journal. 

Appreciate the authors' effort to imrpove the present manuscript.